



# Evaluating the use of Facebook's Prophet model v0.6 in forecasting concentrations of NO$_2$ at single sites across the UK and in response to the COVID-19 lockdown in Manchester, England

David Topping [1], David Watts [2], Hugh Coe [1], James Evans [3], Thomas J. Bannan [1], Douglas Lowe [4], Caroline Jay [5], and Jonathan W. Taylor [1]

[1]Department of Earth and Environmental Science, The University of Manchester, Oxford Road, Manchester, M13 9PL
[2]Transport for Greater Manchester, 2 Piccadilly Place, Manchester, M1 3BG
[3]Department of Geography, The University of Manchester, Oxford Road, Manchester, M13 9PL
[4]ResearchIT, The University of Manchester, Oxford Road, Manchester, M13 9PL
[5]Department of Computer Science, The University of Manchester, Oxford Road, Manchester, M13 9PL

**Correspondence:** David Topping (david.topping@manchester.ac.uk)

**Abstract.** Time-series forecasting methods have often been used to mitigate some of the challenges associated with deploying chemical transport models at high resolution for use at local scales. In this study we deploy and evaluate Facebook's Prophet model v0.6 in predicting hourly concentrations of Nitrogen Dioxide [NO$_2$] over a 2 year period [2018-2019] across the UK's Automatic Urban and Rural Network (AURN). Results indicate promising performance when comparing absolute values,

diurnal trends and seasonality, with discrepancies increasing when the site is classified as having a larger contribution from regional sources and non-local sources. Using mobility and traffic volume data in the model fitting process allowed us to evaluate the ability of the model to forecast levels at two sites in Manchester where there were significant reductions in traffic levels during the COVID-19 lock-down, defined as a national state of restricted access. Prior to lock-down, comparison between hourly concentrations from the Prophet forecast and observations are significantly better compared with predictions from the

EMEP regional model. Despite the simplified approach of fitting to derived NO2-per-traffic volume over a 5 year period, trends in absolute NO$_2$ reductions and diurnal profiles were captured well at Manchester Piccadilly. However at a second site, in Sharston, we found that reliance on historical NO2-per-traffic volume resulted in errors in the prediction as the nature of local traffic changed under the COVID-19 lock-down; correlating with an increase in the Heavy Goods Vehicle fleet [HGV] relative to other forms of traffic. Ancillary meteorological information and predictions from the EMEP model enabled identification of

significant contributions from regional sources during the lock-down period. These periods coincide with noticeable differences between measured and forecast values from Prophet. Overall the Prophet model offers a relatively effective and simple way to make predictions about NO$_2$ at local levels. The source code to reproduce and expand on the work presented in this paper is made openly available.





## 1 Introduction

The impacts of air quality on both human health and the environment remain important areas of research (Manisalidis et al., 2020; Rückerl et al., 2006; Anderson et al., 2012). For predicting how concentrations of pollutants change, there are now a number of modelling platforms that cover local to national scales (e.g., Silveira et al., 2019; Grell et al., 2004). These include those that are built around chemical and process models that reflect key sources, sinks and trans-formative processes, or are based on statistical or machine learning representations fit to empirical observations (e.g., Rybarczyk and Zalakeviciute, 2018).

The ability to forecast levels of pollution is, of course, important to a wide range of stakeholders; not least regional authorities who might want to implement adaptive traffic management strategies that minimise exposure of vulnerable groups. With the rise of internet-of-things [IoT] enabled devices, there has been growing debate around the usefulness of single point and distributed networks of sensors with varying provenance and fidelity (Lewis and Edwards, 2016), and methods that can utilise this information for 'hyper local' forecasting (Huang and Kuo, 2018).

Time-series forecasting methods have often been used to mitigate some of the challenges associated with deploying chemical transport models at high resolution for use at local scales. This does not remove the usefulness, or importance, of the latter methods in understanding wider source contributions and fate, but offers an alternative method for utilising historical local forecasts, and perhaps adapting to locally-driven forces that are important in understanding concentration and composition change. In addition, the boom in smart cities has resulted in often substantial investments in infrastructure for capturing a

large number of ancillary data, which in theory can be useful for understanding changes in air-quality and potential routes for reducing exposure. Incorporating that data into time-series forecasting could enable the aforementioned stakeholders to develop systems for evaluating a series of interventions (Huang and Kuo, 2018).

There have been a number of studies developing and evaluating the use of time-series forecasting tools for air-quality. These include recent demonstrations of Long Short-Term Memory [LSTM] methods, and new demonstrations of combining

Convolutional Neural Networks (CNN) and Long Short-Term Memory (LSTM) methods applied to the PM2.5 forecasting (Huang and Kuo, 2018; Liu and Chen, 2020; Zhu et al., 2017; Khan et al., 2020; Bai et al., 2019). The availability of machine learning and statistical methods, as delivered through common programming languages, has improved significantly over the last decade. This includes widely used package such as Scikit-Learn Pedregosa et al. (2011) and Keras to name a few. From a domain scientist's perspective, having the ability to prototype and tune any new method for forecasting, without requiring

extensive training in building model architectures, is a bonus. These methods are similar in nature to the weather normalisation techniques developed by Grange and Carslaw (2019) and others.

In this study we apply and evaluate the use of a commonly used open-source time-series forecasting model, Prophet (Taylor and Letham, 2018), in forecasting concentrations of $NO_2$ at sites across the UK. As we discuss in the following text, this includes evaluating the use of incorporating data on traffic volume, supplemented by information on traffic type, to evaluate

the ability to capture variations during the COVID-19 global pandemic. Overall the Prophet model offers a relatively effective and simple way to make predictions about $NO_2$ at local levels.





## 2   Methodology

Here we briefly describe the workflow for capturing the data from the Air Quality sites across the UK, fitting and applying the Prophet model and the EMEP regional model.

### 2.1   Prophet


Prophet is a procedure for forecasting time series data based on an additive model where non-linear trends are fit with yearly, weekly, and daily periodicity. Developed for Facebook by Taylor and Letham (2018), it has found applications across various domains, not least driven by the underlying rationale to develop a modular regression model with interpretive parameters that can be intuitively adjusted by analysts with domain knowledge about the time series (e.g., Makridakis et al., 2018; Žunić et al.,

2020; Navratil and Kolkova, 2019). In this study we first use the 'vanilla' version 0.6 model variant in fitting to historical data, provided at hourly resolution, and evaluate its use in forecasting concentrations of nitrogen dioxide ($NO_2$) a month in advance. The internal cross-validation methods provided by Prophet are used to arrive at a set of performance metrics applied across all sites. Distinctly different from cross validation on standard regression or classification methods, this process includes fitting to historical data over a specified period, providing rolling forecasts 30 days in advance, and then repeating the process over

a moving window of measurement periods. For example, in our study we look at predicting monthly forecasts at one hour resolution over the period 2018-2019. This includes fitting to the previous three years of data to forecast the proceeding month, and then repeating the process by moving the start of the historical fit, and thus forecast, by 15 days. There are parameters one can tune when using Prophet which can improve performance. These include manually defining change-points, or points in time where the trend in measured concentrations changes significantly, and the weight given to those points. We refrain from

performing a detailed sensitivity study of these parameters across all sites for a number of reasons, not least the local knowledge that might be needed to interpret significant interventions across regional authorities that should arise through detected change-points. We do, however, use two sites in Manchester to investigate model sensitivities, including the use of incorporating traffic data to supplement the forecasting potential and the impact of pre-processing the measured data to reduce any skewness in the profile of $NO_2$.

### 2.2   EMEP regional model


The European Monitoring and Evaluation Programme for Transboundary Long-Range Transported Air Pollutants (EMEP) model is a 3-D Eulerian chemical transport model that integrates chemical processes (such as dry deposition and washout, emissions from anthropogenic and biogenic sources, gas-phase chemical reactions) with large-scale transport proceses (Simpson et al., 2012). The model version used for this study is v4.33 (201906). The EMEP model is driven by meteorological fields

taken at 3-hourly intervals from the Weather Research and Forecasting (WRF) model (v4.1.3) simulations (Skamarock and Huang), which in turn are driven by ERA-5 global meteorological data (Service).

The model setup consists of two domains. The traditional 50km resolution EMEP domain covering all of Europe (an area of 8,500 x 6,650 km) is used to create chemical boundary conditions for a second, higher resolution (3km) domain cover-





ing the majority of the British Isles (an area of 1,080 x 1,080 km). Anthropogenic emissions for the outer domain are taken

from the TNO database for reporting year 2017, at a resolution of 0.1 x 0.1 degrees. This dataset is provided as a standard input for the EMEP model. For the inner domain, UK National Atmospheric Emissions Inventory (NAEI) emissions (http://naei.defra.gov.uk), for the reporting year 2016 and at a resolution of 1 x 1 km, are used for the UK, Atlantic, and North Sea regions, while TNO emissions are used for the rest of the domain (covering the Republic of Ireland, and parts of France).

The three year (2016–2019) simulation period was split into 18 2-month periods, each preceded by a 7-day spin-up period to

initialise the chemical fields. The 3-month simulation for 2020 (March-May) was run as a single period, with a 7-day spin-up period at the end of Feb 2020.

## 2.3 Data

The Air Quality data is taken from the Automatic Urban and Rural Network (AURN), hosted by the Department for Environment Food Rural Affairshttps://uk-air.defra.gov.uk/networks/network-info?view=aurn. The AURN is the UK's largest

automatic monitoring network and is the main network used for compliance reporting against the Ambient Air Quality Directives. As can be found on the AURN web-page, this network provides measurements of oxides of nitrogen (NOx), sulphur dioxide (SO2), ozone (O3), carbon monoxide (CO) and particles (PM10, PM2.5), depending on the year and site studied. The downloaded data are also combined with meteorological data on wind speed, direction and temperature provided by the UK Met Office. The code used for downloading this data for all sites is discussed in section 4 and delivers parameters at hourly

resolution. In the model fitting we use concentrations of NO2 and all meteorological variables by default. Sites in the AURN network are classified into a small number of types to reflect the pollution environment in the vicinity of the sampling location. These are, briefly, combinations of the following categories listed in Table 1.

In this study, we evaluate data from the following station types between the years 2015 to 2020, with the number of sites given next to each category in Table 2.

Traffic data for this study is provided by Transport for Greater Manchester, the local government body responsible for delivering Greater Manchester's transport strategy and commitments (https://tfgm.com/about-tfgm). Through use of their cloud-hosted data platform, C2 (https://www.drakewell.com/), data from their estate of Intelligent Transport Systems (ITS) could be readily accessed. Included in this estate, are a network of Automatic Traffic Count (ATC) sites, monitoring vehicles using Induction Loop Detectors (ILDs) in the carriageway, which provides vehicle flow, speed and classification data in real-time to

the platform. Approximately 120 sites constitute for this network of ATC sites, strategically located across main routes and the Key Route Network (KRN). Also included as part of TfGM's ITS estate, are a network of approximately 550 Bluetooth Journey Time Passive Sensors, primarily located at Traffic Signal junctions along major routes across the wider Greater Manchester (GM) region. Sensors use an external antenna to capture Bluetooth enabled devices, storing their unique Media Access Control (MAC) address, which is then randomised and encrypted. MAC addresses are collected across the network with an associated

timestamp, allowing journey times to be generated across the Highway Network, utilising the C2 platform to filter outliers. A validation exercise completed internally at TfGM estimates that this method captures between 8 % – 15 % of vehicles making a journey between two sensors, depending on the route.





This means that there is an error in traffic volume inferred from these measurement points, though we demonstrate in section 3.2 that using this metric of mobility, to infer a relative traffic volume contribution to levels of $NO_2$, captures the decreasing trend in measured $NO_2$ well. For the purpose of this study we use data from monitoring sites as close as possible to the Manchester Piccadilly and Sharston AURN sites. We supplement the analysis for Sharston with data on vehicle types from an ATC [Automated Traffic Counting] measurement node.

## 2.4 COVID-19

The UK Government responded to the COVID-19 pandemic with a 'contain, delay, research and mitigate' strategy (Health Foundation, 2020). On 12th March the response moved from contain to delay, with the the announcement of social distancing measures for people with COVID-19 symptoms, and on 16th March people were advised to 'stop non-essential contact with others and to stop all unnecessary travel'. Schools, bars, restaurants, non-essential shops, leisure activities and many work premises were told to close from 20th March. There has since been a gradual relaxing of measures, with people being encouraged to return to work from 13th May, and non-essential shops allowed to open on 1st June. Bars and restaurants were able to open on 4th July. We refer to this national state of restricted movement as lock-down.

## 3 Results

### 3.1 Site analysis for years 2018 and 2019

Figure 1 displays the mean percentage deviation when comparing measured and predicted hourly concentrations of $NO_2$ over a forecast window of 1 month, as a function of AURN site type, when fit to 3 years of historical data. Over the years 2018 and 2019 this covers 35K data-points for comparison at each site. Whilst the profile for each site type can have multiple modes, the smallest error is found at Urban Traffic sites, followed by those classed as Urban Background. As shown in Table 1, these sites dominate the number studied, providing data from 85 sites out of a total of 114. The errors associated with Rural Background peak at near 40 percent difference, as compared with just under 20 percent for Urban Traffic sites. Despite the difference in the number of sites used, this may be a result of the implicit capture of local influences on the measured NO2 signal, predominantly from traffic, as compared with both Urban and Rural Background which might have a larger relative contribution from a range of both local non-traffic and regional background sources. There are only two Suburban Industrial sites, where results reveal two distributions offering some insight into the range of errors for a single site versus the error between sites. These statistics were generated using the absolute value of predicted NO2, but Prophet also provides a forecast uncertainty range. This is generated from uncertainty in the trend, uncertainty in the seasonality estimates, and additional observation noise. Figure 2 displays the percentage of observations that were within model uncertainty bounds, with all sites having a maximum close to 80 percent. As previously noted, there seems to be a variable performance according to site type, and Figure 3 displays predicted versus measured values for all observations at two individual sites. The first (Figure 3a) shows data from an Urban Traffic location in London [Maryleborne Road], the second a Rural Background site in Salford (Figure 3c). Figures 4-5 display





a comparison between Prophet forecasts, measured data and EMEP predictions as diurnal box-plots, using all of the hourly
data from 2018-2019. The Prophet forecasts capture the diurnal trends in each site, whilst the median points demonstrate the
tendency for some level of over-prediction and the inter-quartile range shows the relatively tighter distribution of values from
the forecasts. As has already been stated, understanding the performance of individual sites requires local knowledge, but
these results demonstrate promising capability. At the Piccadilly site, EMEP predictions are consistently lower than measured
values, with discrepancies increasing during the middle of the day. The same is true at the Sharston site, though discrepancies
are reduced. This helps to provide context in section 3.2 when comparing values during the COVID19 lockdown.

The multi-modal behaviour of percentage deviation by site type may be indicative of local interventions not captured by the
default change-points used during the fitting process. Specifically, Prophet is designed to detect change-points of the sampled
trend in observations. One can alter the weighting given to such changes. A manual analysis on an individual site level might
identify significant changes in local activity that would be expected to change the seasonality in measured $NO_2$ and thus define
change-points that need to be captured during the fitting process. We also see in Figure 3 that predictions tend to follow a
normal distribution, which leads to a higher deviation between measured and predicted values when the measured values are
skewed. To investigate potential improvements from giving higher weighting to automatically detected change-points and pre-
processing of the target variable, Figure 6 displays the change in forecast performance when applying combinations of both.
The 'Vanilla' results, from the Vanilla variant of prophet, (CP=10) show predictions when assigning an arbitrary change point
prior scale of 10. PT and PT(CP=10) represent predictions when applying a power transformer to the NO2 data prior to fitting,
with and without a change point prior weighting increase. Power transforms are a family of transformations that are applied
to make data more Gaussian-like. In this instance we use the Yeo-Johnson transformer in the Scikit-learn package and then fit
Prophet to the scaled data, before inverting the transformer to arrive back at an estimated concentration of $NO_2$. In Figure 6(a)
we see the PT(CP=10) model variant has a smaller mean deviation of ∼10% compared with the ∼18% of the Vanilla variant.
Likewise we see that this variant produces a distribution of values that better match the measured values. However we still
see that the predicted distribution is narrower than the measured values, leading to over-estimation at lower concentrations and
under-estimation at higher concentrations.

For any given location, attributing the change in measured NO2 to changes in traffic volume, for example, is implicitly
captured in the use of detected change-points during the fitting process. From an urban planning perspective, being able
to forecast expected changes in NO2 as a result of significant traffic interventions is useful. However, the measured signal
implicitly captures the contribution from local and regional backgrounds, so this requires a method for including traffic data as
an additional variable during the fitting process. In the following section we study changes measured in air quality and traffic
during the COVID-19 lock-down in Manchester.

## 3.2  Incorporating traffic data and response to COVID-19

There have been many approaches used to estimate the fractional contribution to $NO_2$ from traffic within forecasting techniques
(e.g., Carslaw and Beevers, 2005). As Zhang et al. (2013) note, the proximity to roadways and traffic levels are sometimes used
as proxies since, in general, $NO_2$ levels decline with distance from a highway. Other methods include statistical interpolation





(e.g., Jerrett et al., 2007), line dispersion models (e.g., Bellander et al., 2001) and integrated emission-meteorological models (e.g., Frohn et al., 2002). In this study we use the simple approximation of using Prophet to predict the ratio of measured

$NO_2$-per-traffic volume, assuming traffic from an individual site is representative of the dominant source, allowing us to then use the traffic volume data to calculate changes in $NO_2$ directly. This simple approach relies on a number of assumptions, including the ability of Prophet to detect significant changes in trends from both historical interventions and changes in source profiles from local and regional sources of $NO_2$. In the following section, we demonstrate the usefulness of this approach when focusing on two sites in Manchester, UK before and during the COVID-19 lockdown, using traffic data provided by Transport

for Greater Manchester (TfGM).

     The two AURN sites chosen, Sharston [UK-AIR ID: UKA00617] and Manchester Piccadilly [UK-AIR ID: UKA00248], are both classified as Urban Background sites. The Manchester Piccadilly site is located in Manchester City Centre. Whilst the nearest major road and traffic monitoring site [A5103, Portland Street] is 200m away from the site, the site is also located close to a major bus hub and adjoining road networks. The Sharston site is located 40m from the nearest road, and traffic monitoring

site, and 1.5km from the edge of a runway at Manchester Airport. The data from Portland Street was used to extract $NO_2$-per-traffic volume used in the fitting process for the Manchester Piccadilly site. As we state in section 2.3, the Bluetooth Journey Time sensors are subject to some error when interpreting vehicle traffic volume. Nonetheless, we evaluate the use of said data in the fitting process and assume this mobility metric is representative of local traffic. Following Piccadilly, the Sharston site used traffic volume data 40m away to fit the $NO_2$-per-volume data, whilst vehicle type data from an ATC site ∼1.7km away

was also used to interpret results, assuming that site was indicative of behaviour of the broader industrial and transport hub.

     Figures 7(a) and 7(b) display the hourly volume of traffic near to both sites [top rows] and the inferred natural log(NO2-per-traffic volume) by combining this information with measurements of $NO_2$. In both cases the significant decrease in traffic can be observed in late March, with a noticeable gradual increase in the Sharston data. The log($NO_2$-per-traffic volume) inferred from the Piccadilly site appears to be less noisy than that at Sharston, with an increase in the daily range occurring after the

COVID-19 lockdown.

     Figure 9 (a) displays the predictions from Prophet without incorporating the traffic data [blue lines] and incorporating the traffic data [green lines] at Manchester Piccadilly. The measurements are shown by the red lines. We also show the predictions from the EMEP model. As the date approaches the COVID-19 lock-down, the measured values significantly decrease. The blue lines provide an indication of forecasts assuming a 'business as usual' situation. One would also expect this from the EMEP

predictions though we have seen in figure 4, that EMEP predictions are systematically lower than measured values at this site, especially during the day. The green line captures the trend and seasonality in the measured $NO_2$ signal very well. This is confirmed in Figure (a) where the diurnal profile of measured values and both sets of forecasts are given for dates after lockdown. One can clearly see the significant decrease across all days, when referencing 'Measured' values against the 'Forecast' values. The incorporation of traffic data into the forecasts significantly improves the predictions of the diurnal trends, though

the distribution is much tighter than the measured values. Again, the EMEP predictions are surprisingly good, though we have seen the systematic under-prediction at this site. There are periods of significant increases in $NO_2$ not captured by the Prophet forecasts in the time series plots in figure 9(a), that are captured from the EMEP predictions. Figure 8 displays a windrose of





percentage error between the Prophet predictions, using traffic data, and measured NO$_2$ concentrations at both Piccadilly [left figure] and Sharston [right figure].

The significant increases in measured NO$_2$ in late March, as well as early- and mid- April, are attributable to periods of slack winds and easterly flow from continental Europe. We investigated the variation in air-mass type throughout this period, by running 72-hour back trajectories using the Hybrid Single Particle Lagrangian Integrated Trajectory Model (HYSPLIT) v4.2.0 [Draxler and Hess, 1998]. Three-hourly backtrajectories were initiated ∼500m above from a release point in south Manchester, roughly midway between the Manchester Piccadilly and Sharston AURN sites. Bearing in mind the uncertainty

associated with individual trajectories, we used the Ward's method clustering described by Stunder (1996) to classify air-mass origin into broad regions, rather than to try identify specific sources. The spatial variance of each cluster pair is calculated, defined as the sum of the squared distances between the end points of the potential new cluster's component trajectories and the mean of the trajectories in that cluster. The pair with the lowest spatial variance is merged into a new cluster with each iteration. As the clusters grow larger with each iteration, the total spatial variance (the sum of the spatial variance of all clusters)

grows larger, and eventually reaches a point where it increases quickly as dissimilar clusters are merged, reaching a maximum as all clusters are merged into one. The ideal number of clusters is just before this sharp increase; in this case 6 clusters.

Figure 10 shows the results of the clustering technique in terms of the broad region of each cluster and a time series of which cluster trajectories were assigned to. For most of March, the clusters from the north Atlantic and North sea were dominant, and these were associated with clean background air-masses. Periods in late March, early April and mid April were dominated by

more easterly and southerly air masses, with slack winds common (represented here by the 72-hour back-trajectory end points not being as far from the source). The slack, southerly winds in the cluster from the southern UK were particularly associated with higher levels of NO$_x$ in early April.

Figure 9 (b) displays the hourly data from the Sharston site. In this case, whilst a decrease in measured NO$_2$ is seen, this is smaller than the decrease in NO$_2$ observed at the Piccadilly site, despite the decrease in traffic volume inferred from the nearby

site. This is also confirmed in Figure 11 (b) which shows a marked under-prediction of concentrations in the early hours and towards late evening. To investigate this further we analysed data from an ATC site located ∼1.7Km away from the AURN and original traffic site. This site, whilst further away, provides data on segregated vehicles types. In Figures 12(a) and (b) we plot the diurnal profile of Heavy Goods Vehicles [HGVs - top plot in 12(a)] and non-HGV vehicles [top plot in 12(b)] along with the change in relative contribution of HGVs to total traffic both before [blue boxes] and after lock-down [orange boxes].

In these plots we see a significant shift in the fractional contribution of HGVs in the early hours. It is plausible this may be one reason behind the under-predictions of NO$_2$ concentrations when fitting to inferred historical trends in NO$_2$-per-traffic, though we repeat we are assuming this site is representative of the local traffic near the AURN site and using the relative higher emission factors of NO$_2$ from HGVs (Carslaw and Rhys-Tyler, 2013).





## 4  Conclusions and future work

In this study we conduct an analysis of using Facebook's Prophet model in predicting hourly concentrations of Nitrogen Diox-
ide [$NO_2$], over 30 day forecasts, over a 2 year period [2018-2019] across the UK's Automatic Urban and Rural Network
(AURN). Results indicate promising performance when comparing absolute values, diurnal trends and seasonality, with dis-
crepancies increasing when the site is classified as having a larger contribution from regional sources and non-local sources.
Focusing on two sites in Manchester, we find the use of transport data in the fitting process could capture the measured decrease
in $NO_2$ well during the COVID19 lock-down. Results at these sites also demonstrate improved performance over the standard
regional model EMEP during 2018-2019, where some improvements in performance can be found by changing change-point
prior scales and pre-processing the measured data. Discrepancies between forecasts that incorporate traffic and measured val-
ues could arise from errors associated with a number of factors, but data on vehicle traffic type suggests this could also be due
to an increase in the ratio of Heavy Goods Vehicles [HGVs]. Despite the systematic error associated with EMEP predictions,
combining regional and time-series forecasts help identify periods of regional background influence that might be otherwise
hidden in the inferred local traffic contributions in normal conditions.

     There are a number of potential improvements that could be made in taking this work forward. As we already note, un-
derstanding the impact of any previous interventions at a local level might help better understand the reported errors. The
study applied to Manchester might be replicated elsewhere, should traffic data be available. We note the errors associated with
journey time to traffic volume inference. We have not used data on boundary layer height, or ancillary activity data during the
fitting process. However ongoing work is looking at the value of incorporating a range of data products.

     Following the rationale behind its design, the Prophet model offers a relatively simple and effective solution for domain
scientists, local authorities and transport authorities to predict the impact of measures to reduce traffic on air quality. The open
access nature of the Prophet model plus the increasing availability of traffic data from transport authorities makes this kind of
prediction possible across many towns and cities. It also offers a local level resolution for NO2 predictions that has been hard
to achieve in the past, but which is essential for decision-makers seeking to reduce traffic and improve air quality in urban areas
through geographically targeted interventions.

*Code and data availability.*  In this paper we combine scripts to download the AURN data and then fit/evaluate the Prophet model to said
data. The current version of these scripts are available from the project website https://github.com/loftytopping/Prophet_forecasting_AQ
under the licence GPL v3.0. The exact version of the scripts used to produce the results used in this paper are archived on Zenodo
(https://zenodo.org/record/3978645). This includes the EMEP generated and transport data from TfGM. For queries regarding additional data
requests from TfGM please contact David Watts, david.watts@tfgm.com. This repository also contains a Conda environment .yml file for
replicating the collection of packages needed to repeat the analysis, including the Prophet model v0.6. The project website for the Prophet
model can be found at https://facebook.github.io/prophet/. Version 0.6 was used in this study. The project website for the EMEP model can
be found at https://github.com/metno/emep-ctm. Version 4.33 2019 was used for this study. The operational scripts for the exact setup of





EMEP can be found on Zenodo (https://zenodo.org/record/3997301), as can the input files (https://zenodo.org/record/3997271) and configuration files (https://zenodo.org/record/3997165).

*Author contributions.* D Topping wrote the scripts for the Prophet model fitting, AURN data download and model evaluation. J Evans and T Bannan are the PI and senior researcher of the Manchester Urban Observatory respectively and jointly manage the collaboration with TfGM
for access to traffic data. David Watts is an ITS Engineer for TfGM and provided us with access to the transport data-hub whilst helping us interpret the extracted data. H Coe and J Taylor are the lead PI and senior researcher of the Manchester Air Quality Supersite respectively, providing the back trajectory analysis. D Lowe carried out the EMEP simulations under an Alan Turing Institute funded project for which C Jay is PI and who also contributed to the output interpretation. Everyone helped to interpret the results and contribute to the writing of the paper.

*Competing interests.* No competing interests are present

*Acknowledgements.* This work was supported by the EPSRC UKCRIC Manchester Urban Observatory (University of Manchester) (grant number: EP/P016782/1) and the Alan Turing Institute funded project, 'Understanding the relationship between human health and the environment'. The authors would like to acknowledge the assistance given by Research IT, and the use of The HPC Pool (funded by the Research Lifecycle Programme at The University of Manchester) for the EMEP simulations. NAEI data is used under Crown 2020 copyright Defra
BEIS via naei.beis.gov.uk, licenced under the Open Government Licence (OGL).



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





| Brief description | |
|---|---|
| Site classification | Description |
| Urban area (U) | Continuously built-up urban area meaning complete (or at least highly predominant) building-up of the street front side by buildings with at least two floors or large detached buildings with at least two floors. Urban sites should measure air quality which is representative of a few km2. |
| Suburban area (S) | Largely built-up urban area, reflecting contiguous settlement of detached buildings of any size with a building density less than for 'continuously built-up' area. Suburban sites should measure air quality which is representative of some tens of km2. |
| Rural area (R) | Sampling points targeted at the protection of vegetation and natural ecosystems shall be sited more than 20 km away from agglomerations and more than 5 km away from other built-up areas, industrial installations or motorways or major roads, so that the air sampled is representative of air quality in a surrounding area of at least 1 000 km2. |
| Traffic station (T) | Located such that its pollution level is determined predominantly by the emissions from nearby traffic (roads, motorways, highways). Sampling probes shall be at least 25 m from the edge of major junctions and no more than 10 m from the kerbside. |
| Industrial station (I) | Located such that its pollution level is influenced predominantly by emissions from nearby single industrial sources or industrial areas with many sources. Air sampled at industrial sites is representative of air quality for an area of at least 250 m × 250 m. |
| Background station (B) | Located such that its pollution level is not influenced significantly by any single source or street, but rather by the integrated contribution from all sources upwind of the station. At rural background sites, the sampling point should not be influenced by agglomerations or industrial sites in its vicinity, i.e. sites closer than five kilometres. |

**Table 1.** Description of site types covered by the AURN network.

| Number of sites studied | |
|---|---|
| Site type | Number |
| Urban Background | 48 |
| Urban Traffic | 47 |
| Rural Background | 14 |
| Urban Industrial | 9 |
| Surburban Background | 4 |
| Suburban Industrial | 2 |

**Table 2.** Number of site types analysed in our study.





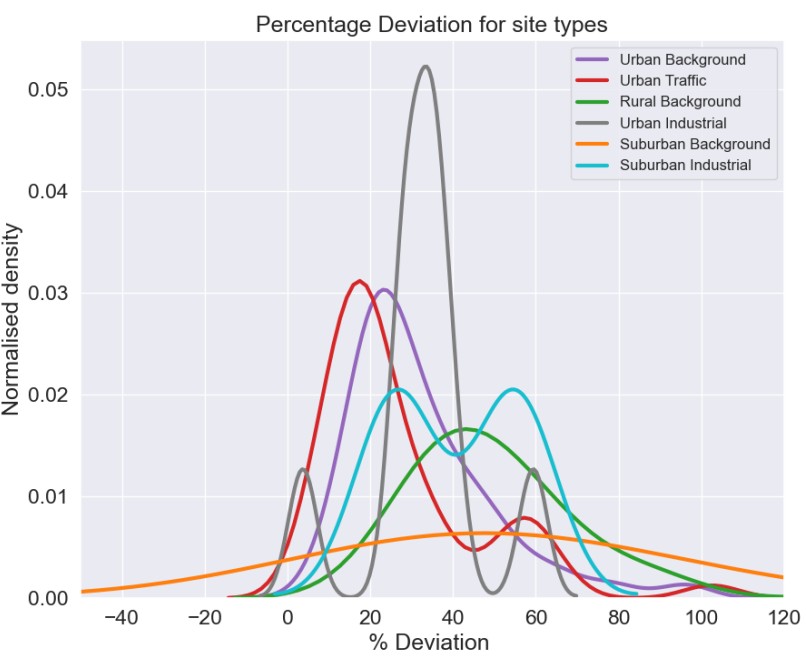

(a) Percentage deviation

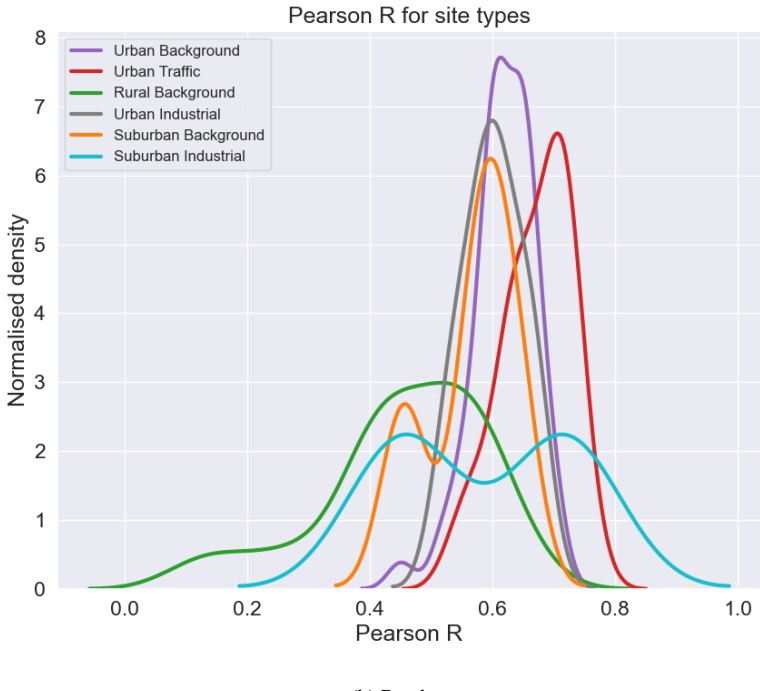

(b) R value

**Figure 1.** Kernel Density Estimate (KDE) of the mean percentage deviation between predicted and measured concentrations of NO$_2$ as a function of site type (a) with subsequent distribution of mean Pearson R values (b))



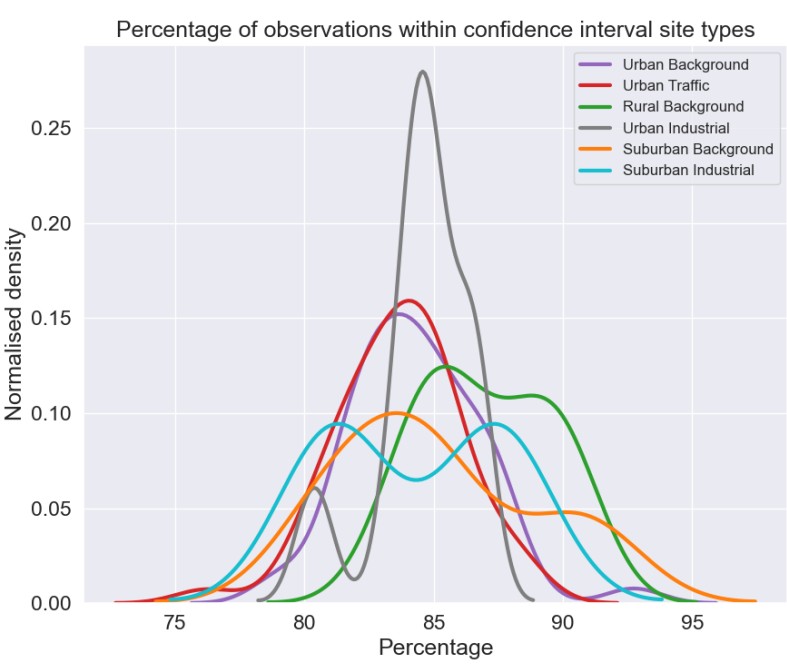

**Figure 2.** Kernel Density Estimate (KDE) of the percentage of observations of NO$_2$ that fall within the forecast uncertainty bounds, as a function of site type

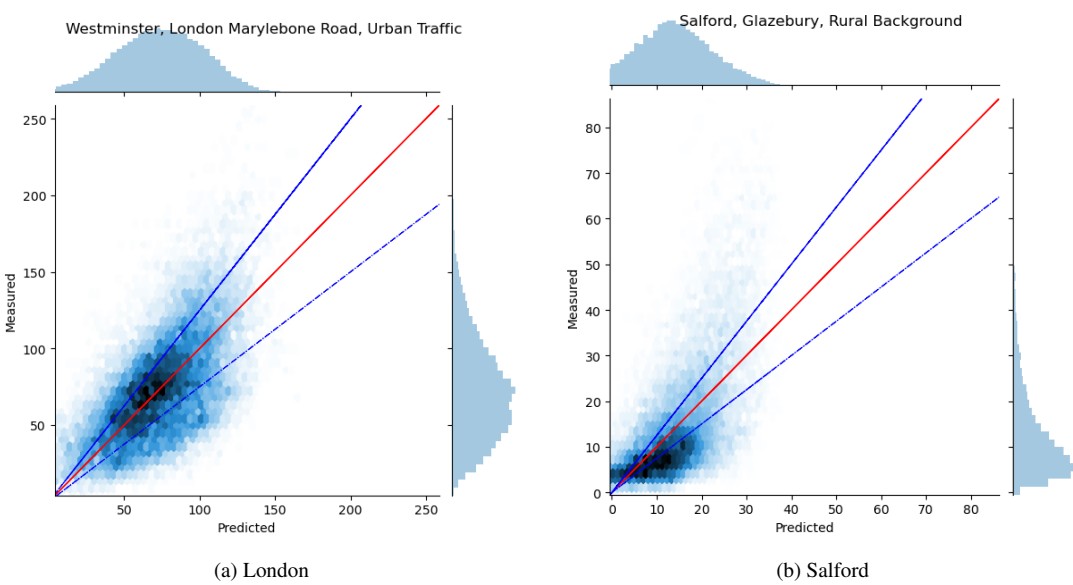

(a) London                                          (b) Salford

**Figure 3.** Hex plots of hourly concentrations of NO$_2$ forecast at 2 sites, representing Urban Traffic (a) and Rural Background (b). The red line represents the 1:1 relationship, whilst the red lines 25 percent above and below

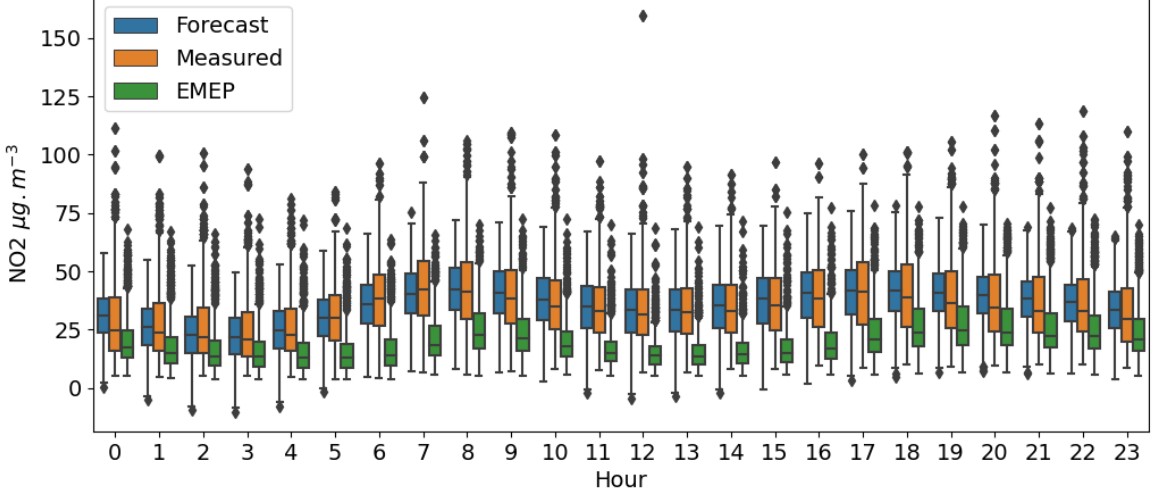

**Figure 4.** Diurnal profile of hourly forecasts over a 30 day period, over 2018 and 2019, at Manchester Piccadilly. The orange boxes represent measured values, the blue boxes predicted.





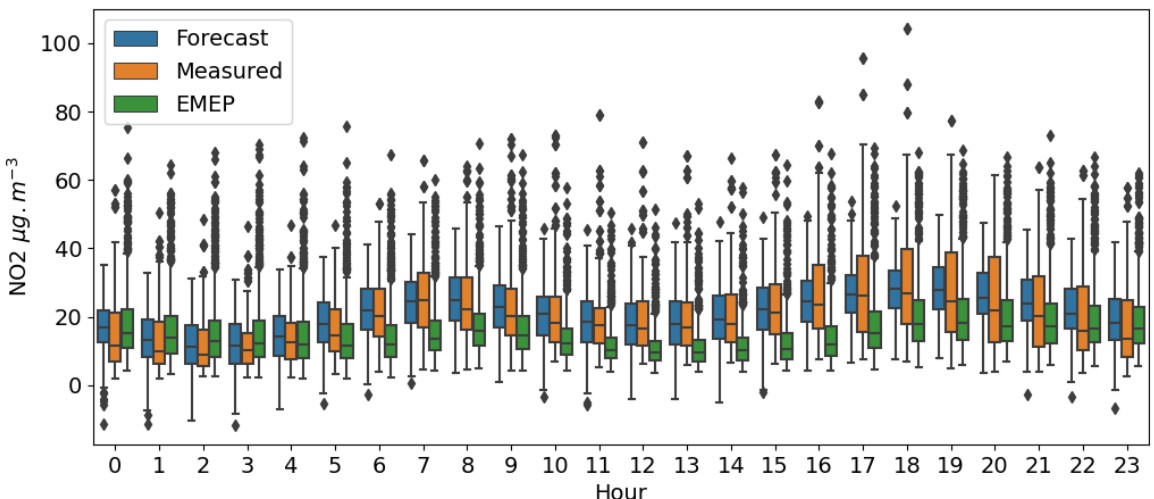

**Figure 5.** Diurnal profile of hourly forecasts over a 30 day period, over 2018 and 2019, at Sharston, Manchester. The orange boxes represent measured values, the blue boxes predicted.

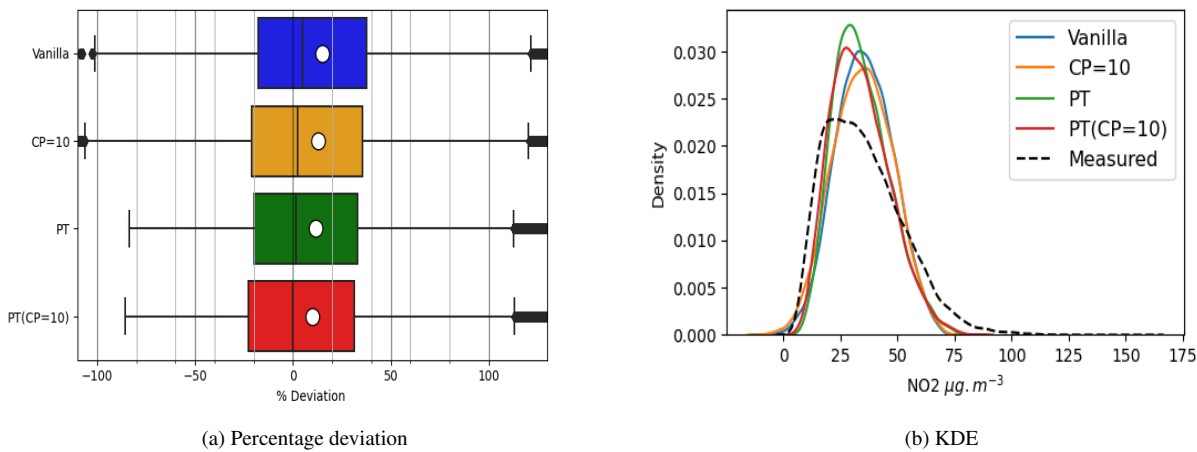

(a) Percentage deviation

(b) KDE

**Figure 6.** Percentage deviation dependency on data pre-processing and weighting of change-points as box-plots (a) and KDE distributions (b) for predictions at Manchester Piccadilly. In the box-plots the white circles display the mean values





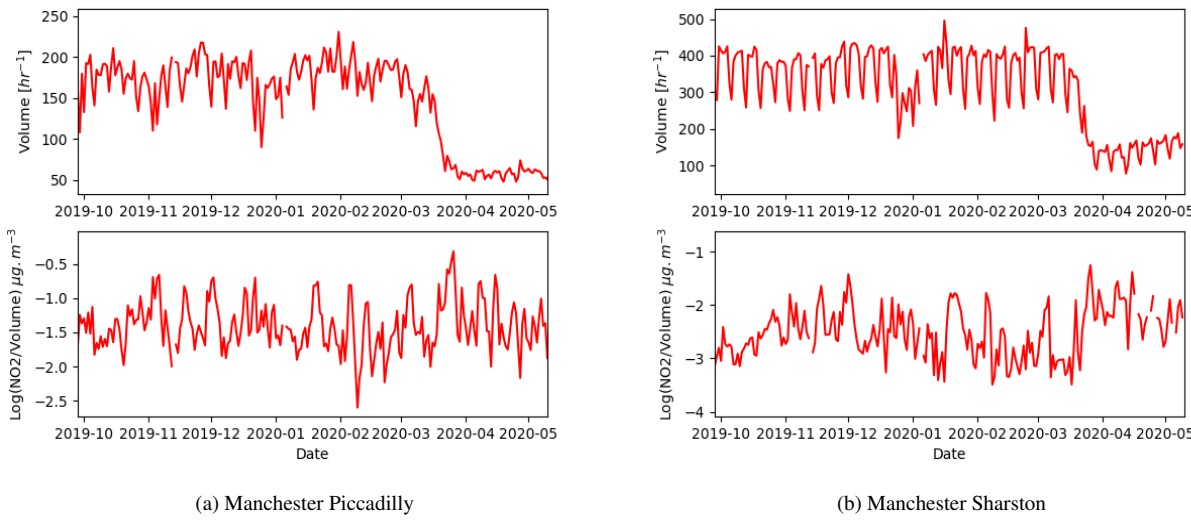

(a) Manchester Piccadilly

(b) Manchester Sharston

**Figure 7.** Change in daily mean traffic volume [top row] with inferred log(NO$_2$/Volume) [bottom row] by combining the AURN and traffic data at Manchester Piccadilly (a) and Manchester Sharston (b)

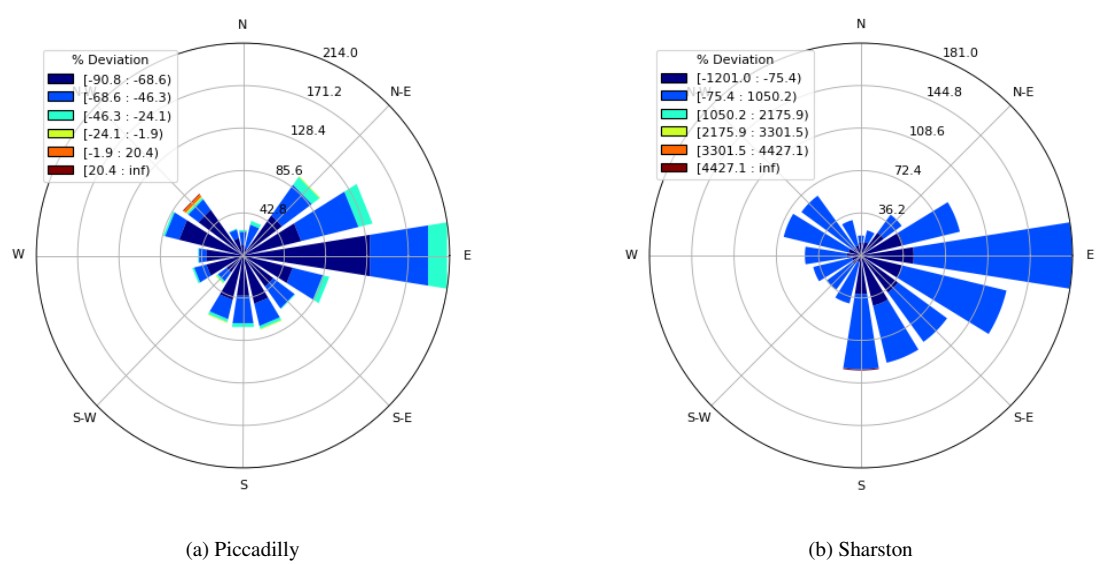

(a) Piccadilly

(b) Sharston

**Figure 8.** Percentage deviations by wind direction, during the COVID-19 lockdown period, for both Piccadilly [left plot] and Sharston sits [right plot]

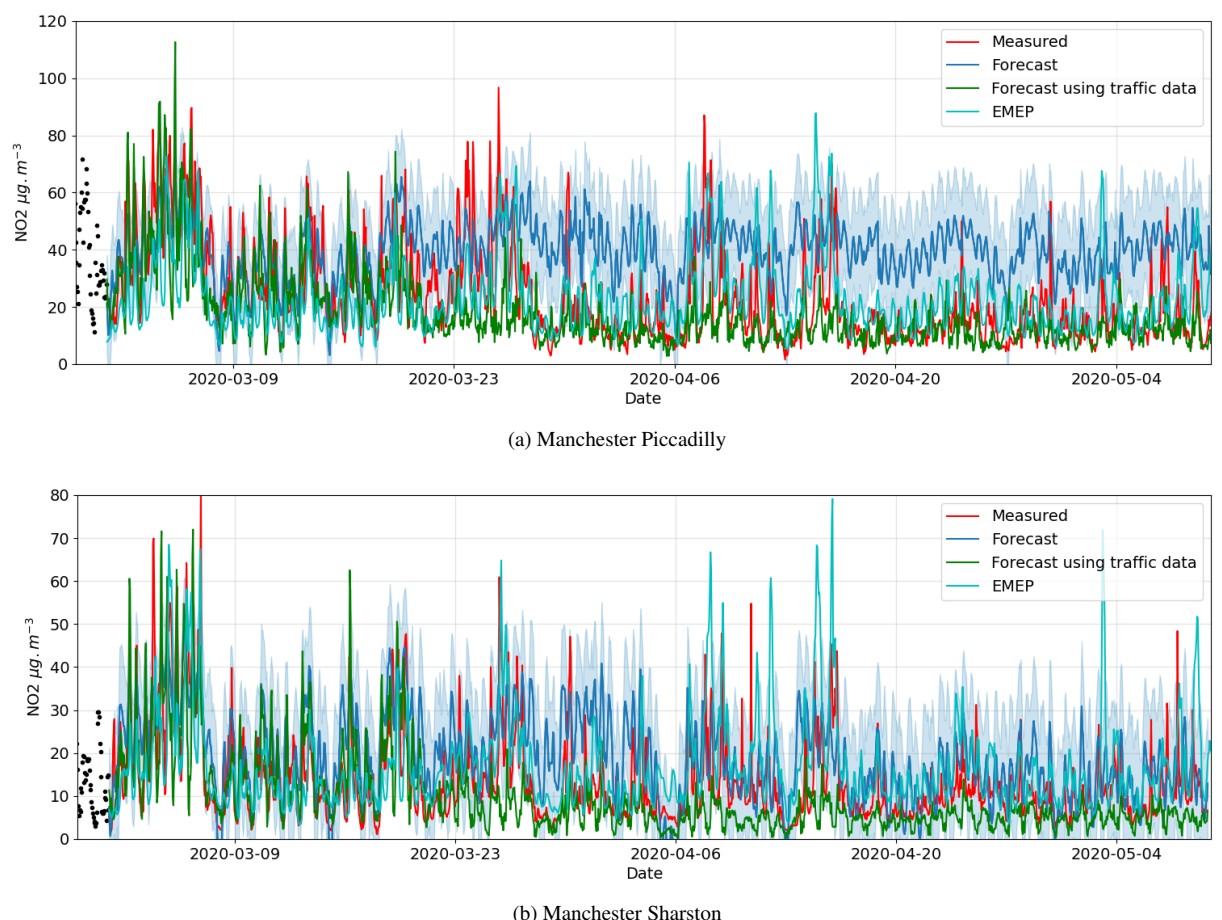

(a) Manchester Piccadilly

(b) Manchester Sharston

**Figure 9.** Actual NO2 [red line] versus predicted concentrations assuming 'business as usual' [blue lines] and based on predicting log($NO_2$/volume) [green lines] for Manchester Piccadilly (a) and Manchester Sharston (b)

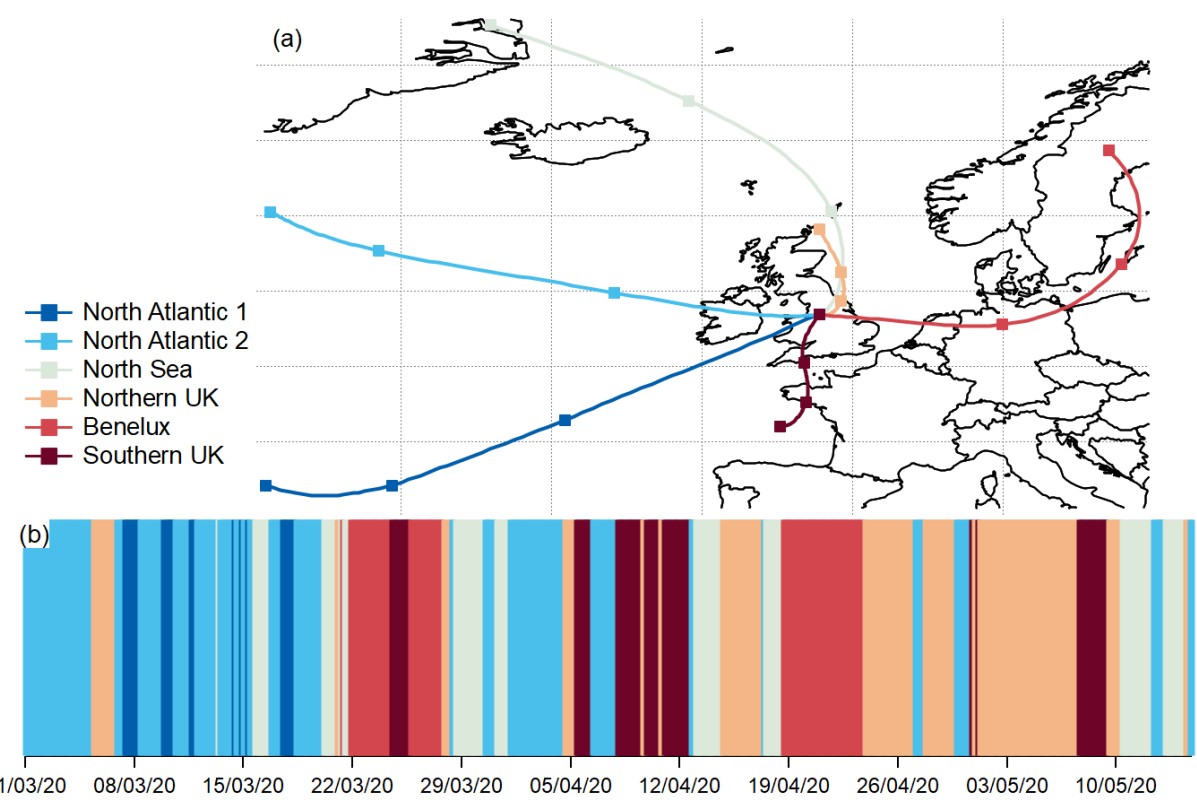

**Figure 10.** Clusters of 72-hour back-trajectories. Panel (a) shows the average position of each cluster, with markers every 24 hours, and panel (b) shows the time series of each cluster. The center lines show the mean position of all the trajectories in the cluster at any given point in time, and the reader should be aware that these represent the centre of a broad region of air-mass origin, rather than a path to an exact source location.

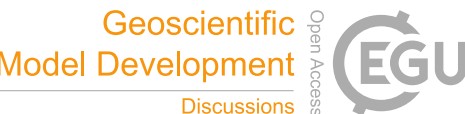

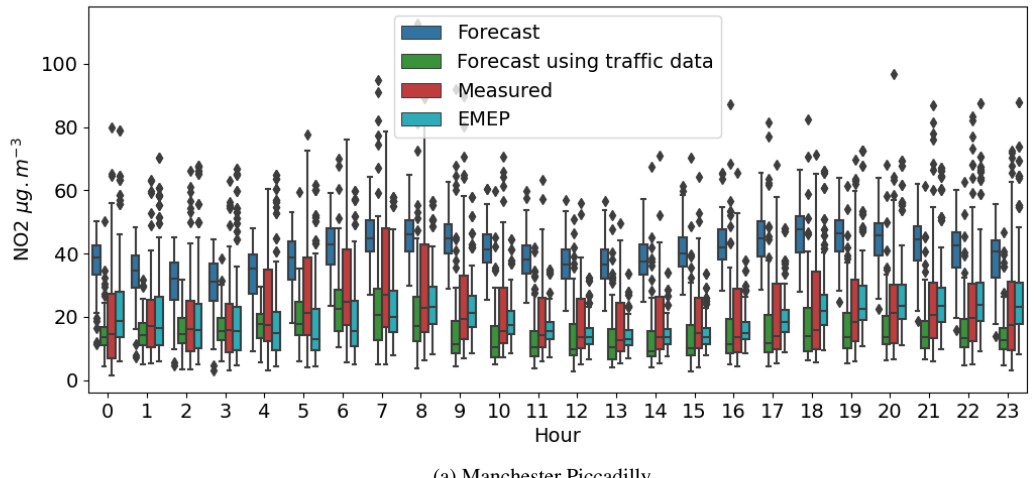

(a) Manchester Piccadilly

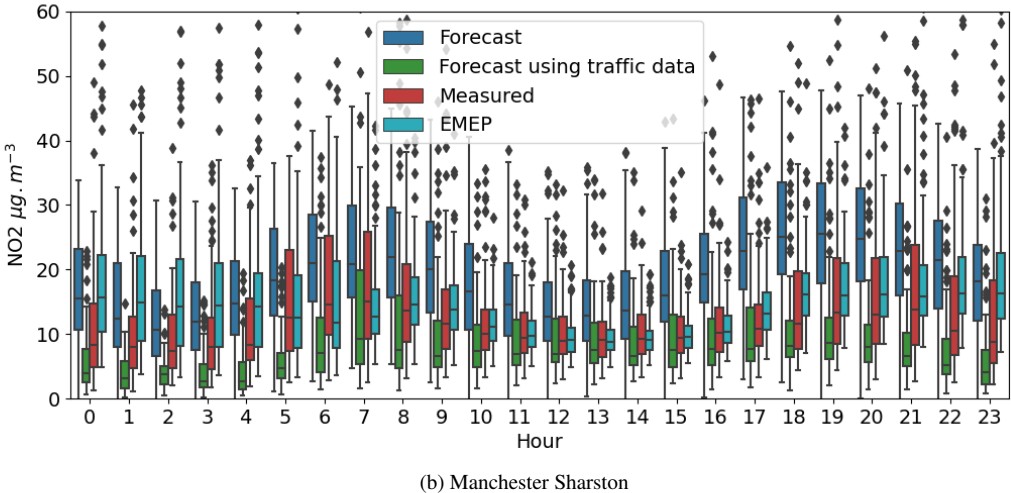

(b) Manchester Sharston

**Figure 11.** Comparison of diurnal profiles, after the COVID-19 lock-down, between measured values [Blue], predicting $NO_2$ from historical data [Orange] and predicting Log(NO2/Volume) to arrive at $NO_2$ [Green] for Manchester Piccadilly (a) and Manchester Sharston (b)



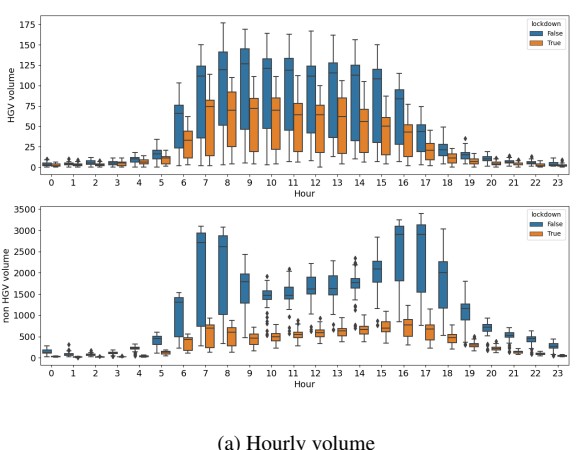

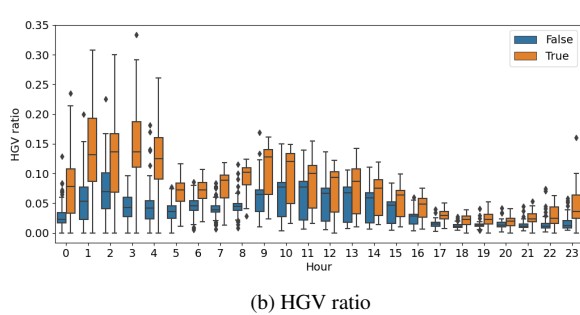

(a) Hourly volume            (b) HGV ratio

**Figure 12.** Change in traffic volume by separating HGV and non-HGV vehicles (a) and comparing the relative change in HGV ratio of total traffic (b) near Sharston, before [blue boxes] and after [orange boxes] the COVID19 lock-down