# Peer review of "Evaluating the use of Facebook's Prophet model v0.6 in forecasting concentrations of NO2 at single sites across the UK and in response to the COVID-19 lockdown in Manchester, England"

_Geoscientific Model Development, 2020_

## Referee Comment (RC1) · Anonymous Referee #1 · 21 Dec 2020

Topping et al. use a time series forecasting toolbox 'Prophet' to predict NO2 pollution at various UK measurement sites. While I note that the manuscript might have significant potential both scientifically and methodologically, I find its current value for the scientific community difficult to judge. In particular, the manuscript lacks clarity in its presentation in a number of critical aspects. For instance, there is insufficient detail in the description of the methodological approach and of useful benchmarks for their predictions. I therefore recommend major revisions subject to which I might be able to provide a more informed opinion on the quality of the work presented here. As I said,

the issues might be mostly presentational but I would need to see this confirmed.

Major comment:

- I find the current description of the methodology insufficient, especially considering the journal's modelling scope and the importance of the 'Prophet' procedure for this work. All that is said is that 'Prophet is a procedure for forecasting time series data based on an additive model where non-linear trends are fit with yearly, weekly, and daily periodicity' and then the authors point towards a reference and some 'interpretive' parameters. This is simply not enough. A schematic would help and there has to be additional insight into how the algorithm works. I am also not entirely sure what is meant by 'rolling forecasts 30 days in advance' (L64). Do I understand correctly, that you train and cross-validate on three years and then predict every single hour for the next 30 days in advance before moving the training interval 15 days forward in time? What are the predictors in this procedure? I assume it must be meteorological variables and NO2 concentrations of the previous hour, and lagged weekly, daily, and yearly values (L56/57)? From the entire manuscript I could not fully comprehend what your predictions are actually based on. My initial impression was that you actually try to predict NO2 30 days in advance at some point, which I assume cannot possibly be the case? Does, e.g., the prediction at day 20 in the prediction period at noon, know of the true NO2 value just one hour earlier? Please clarify these aspects.

Point-by-point comments:

- Abstract l4-l6: could you be more quantitative or at least indicate somehow what "promising performance" means? When I read this for the first time, I was also confused what 'regional' sources are compared to 'non-local' sources and why BOTH should affect performance negatively. What would be a good setting in comparison to these two settings?

- L10: I find the 'simplified approach of fitting to derived NO2-per-traffic volume' approach is introduced somewhat surprisingly and without context. Why is this used? Is that part of one of the models, as indicated by the word 'despite' at the beginning of the sentence? Please clarify.

- L16: 'effective and simple' relative to what?

- L21: clarify the timescale of predictions you are interested in. Minutes? Simultaneous (from other measurements)? Hours? Days? As I said, I am not entirely sure yet if you actually intend to predict just the update in NO2 concentrations from one hour to the next, or over longer time intervals.

- L30: Could you give examples of what those challenges are?

- L43: packageS such as Scikit-Learn and bracket typos. Keras has no official citation?

- L51: define what you mean by 'local' – point/site measurements I suppose?

- L61: here you mention for the first time that you aim to predict NO2 concentrations a month in advance (which could be understood in different ways). I think this should be clarified earlier on, even in the abstract.

- General comments on section 2.1: given how central the Prophet model is to your paper, I would want to see a more detailed description of what happens 'under the hood'. Currently, and given that I don't know the method, it seems like a black box to me, which is impression you would certainly want to avoid. A graphic/schematic would help, too. Why should the Prophet model be advantageous over for example LSTMs? Explaining the modeling process would be really important to make the paper more interesting and more accessible.

- L78 typo processes

- L85: TNO database not defined yet. Maybe link website if appropriate?

- L.89-91: without context this doesn't make sense to me. If there are 2-months periods, what happened to Jan/Feb 2020? I assume you treat 2020 separately due to the lock-down? Maybe worth pointing out already at this stage.

- L.103: make clearer how the different time periods align somewhere in the manuscript (certain years are used for training, others for prediction and EMEP modelling, I suppose, others again to test the effects of the lock-down?).

- L.131: 3. Results. By this point in the manuscript, I am still not sure how you predict NO2, but you already start presenting results. What are the variables you use as predictors (and at what time lags)? How do you cross-validate to avoid overfitting (you mention that something is done in section 2.1, but I feel this is insufficient for a modelling paper)? How do you evaluate model skill? Surely the Pearson correlations shown in Figure 1b are performed on test data and not on training data, i.e. the sequential predictions on predicted months? Your current manuscript simply does not describe this in sufficient detail and clarity in my opinion and I find it difficult to judge the skill of your approach as a result.

- Figure 1b: Please compare these correlations to time series where you simply prescribe a seasonal cycle (smoothed over several days) or a constant value that is representative of the true annual mean observed values. This would provide a much clearer impression of how much better your predictions actually are compared to very basic models. Urban sites I would expect to show less relative (in %) seasonal variation so that this might explain your smaller error there. Furthermore, try the R2-score (coefficient of determination) rather than just Pearson correlation. The former should be better suited to compare time series of this kind because it does not just consider variance but also magnitude of the prediction error.

- L153: I am sceptical that the EMEP model is a useful benchmark here. It simply seems to have a high bias, but how about an empirical model that simply has no mean bias (for instance the seasonal cycle model mention above, derived from actual observations)?

- L175: At this point it is still not clear to me which variables were actually included in the predictions of results section 3.1...

- Figure 7: relabel y-axis with logarithmic scale I suggest.

- L185: I think this approach requires a more detailed explanation. How is this done exactly with Prophet?

- L207: it is also not clear to me how the traffic data is included in the model. If it is a forecast, do you take traffic measures from the previous hour? How is this relevant to the current hour?

- L220ff: suddenly there is a lot of methodological detail. You have long lost me by then though.

- L245: possibly, but then it could also just be a failure of the regression model to perform well in an effectively different environment?
* * *

---

## Referee Comment (RC2) · Anonymous Referee #2 · 24 Dec 2020

Overall Comments: The analysis needs to be extended as the whole forecast workflow to understand the human impact on NO2 concentrations and forecasting seems to be procrustean to Facebook's Prophet model.

The work that needs to be done for this contribution is major, as relevant forecast diagnostics need to be included, regardless of whether they are implemented by Facebook or not.

The novelty of the paper is the workflow that involves resolving multiple data sources,

that is then used in forecasting and clustering. The workflow does not exist in the write-up (methodology section).

In addition, exploratory data analysis to drive the type of model used (referred to as vanilla without any mathematical description) needs to be provided. Authors need to provide motivations behind models they select and this is not possible without exploratory data analysis. Missing are general approaches such as partial autocorrelation analysis of time series, methods used for resolving different spatial scales of input data, the type of forecast method used (including the trend and seasonality model), and why Prophet model is chosen.

A new section on exploratory data analysis with concrete mathematical motivations to use a specific model is a must for this paper. Prophet model needs to be explained in the context of the problem.

Comments on Sections:

2.1 needs to be extended. In its current form this section is not informative about the model being used. The prophet model has many models, pre and post processing steps. Models used in the study need to be clearly discussed.

2.2 Similar problem as 2.1 This section contains an overly abridged description of the method followed by details regarding application. This section needs to be rewritten.

The methodology requires a workflow section that describes the overall workflow built for this study.

Table 1 shows differing spatial resolutions, authors need to clearly present how this disparity is resolved (interpolation, sampling, etc.)

2.4 THis section contains no information on data. If the only purpose of this section is state the changes to mobility patterns after March 12th, this can be done in the section on data sources.

GMDD
The prophet model is a general additive model for time series, where seasonanlity, trend, and cyclicity are linearly superimposed. There is no motivation in the paper why such additive model will be appropriate for NO2 concentrations. It is customary to provide at least a partial autocorelation function plot of the signal in question and performing statistical tests on these components to understand whether an additive model would be appropriate to start with.

Specific Comments:

Line 2: Missing comma after In this study Line 12: "the nature of local traffic": artificial constructs do not possess natural qualities. If you mean historic patterns, please write so. Line 13: Why is HGV abbreviation in square brackets? Line 16: Overall missing comma Line 56-60 belongs in the introduction section. Line 62: "The internal cross-validation methods provided by Prophet are used to arrive at a set of performance metrics applied across all sites." This is not specific enough, you need a subsection under methodology for diagnostics and discuss why they are relevant. Line 63-64: 'this process includes fitting to historical data over a specified period", does this imply you are using an autoregressive forecasting method. If so, this does not imply novelty as these methods have been studied intensively for a long time. Line 65 - 74: These descriptions have nothing to do with the Prophet model but the problem definition. Please distinguish the method from the application.

Line 82-91: Here application is discussed, this section is not related to the method. EMEP model is an integral piece of this work, please describe the system being solved with this model here.

'Line 94: need to be carried over to notes or the appendix

Line 98: "meteorological data on wind speed, direction and temperature provided by the UK Met Office." Please elaborate which model, data product, whether it is reanalysis data. In addition, please introduce
Line 116: "this method captures between 8 % – 15 % of vehicles making a journey between two sensors,". Does this mean the traffic data captures up to 15% of actual traffic. If so the sampling bias, and portion of the traffic needs to be discussed in extenso.

Line 156-157: "The multi-modal behaviour of percentage deviation by site type may be indicative of local interventions not captured by the default change-points used during the fitting process." This seems to be a speculation rather than a result. This can be an important point to bring up in the discussions.

Line 157-158: "Specifically, Prophet is designed to detect change-points of the sampled trend in observations."

Line 158 - 160: "One can alter the weighting given to such changes. A manual analysis on an individual site level might identify significant changes in local activity that would be expected to change the seasonality in measured NO2 and thus define changepoints that need to be captured during the fitting process." This is a use tip about Prophet, I don't think this relates to results.

Line 164: Vanilla prophet is not discussed methodologically. This must be expanded in the methodology section.

Line 169: Improvement in forecast from 18% to 10% deviation can be quite misleading. Authors need a transform bias correction scheme for the Yeo-Johnson transform to address this. Please see the seminal work of Beauchamp and Olson (1973) on pitfalls of using transformations. Secondly, a transformation is applied to data without being introduced in the methodology. There is no discussion of why this transformation is chosen over Box-Cox transform or a log transform.

Line 220: HYSPLIT model is introduced for the first time under results. This is a major part of the method yet not discussed in methodology.

Line 225: Ward's Method is introduced for the first time in results without discussion
**under methodology**

Line 227-229: "Discrepancies between forecasts that incorporate traffic and measured values could arise from errors associated with a number of factors, but data on vehicle traffic type suggests this could also be due to an increase in the ratio of Heavy Goods Vehicles [HGVs]." This statement is not possible to make with current metrics and the level of exploratory analysis of variables. Bias residuals, in addition to very common forecast metrics such as MAPE and MASE are missing in the analysis. If NO2 concentrations have strong seasonality, then observing bimodal deviance (such Figure 1 is not surprising, that is just an indication that there are two regimes where model does significantly bad compared to others).

Figure 4: On average, the model consistently over-estimates measured values. This needs to be discussed.

References: Beauchamp, J. J., & Olson, J. S. (1973). Corrections for bias in regression estimates after logarithmic transformation. Ecology, 54(6), 1403-1407.

---

## Author Comment (AC1) · 1 Mar 2021

**Response to anonymous reviewer 1**

David Topping

Dear colleague.

Many thanks for taking the time to conduct a review which was submitted 21/12/20. I am of course very happy to respond to all points raised. In the following text I provide a response to each point raised [formatted in italic].

*General comment: Topping et al. use a time series forecasting toolbox 'Prophet' to predict NO2 pollution at various UK measurement sites. While I note that the manuscript might have significant potential both scientifically and methodologically, I find its current value for the scientific community difficult to judge. In particular, the manuscript lacks clarity in its presentation in a number of critical aspects. For instance, there is insufficient detail in the description of the methodological approach and of useful benchmarks for their predictions. I therefore recommend major revisions subject to which I might be able to provide*

*a more informed opinion on the quality of the work presented here. As I said, the issues might be mostly presentational but I would need to see this confirmed*

**Response:** Apologies for any confusion. In the responses to the proceeding points raised, and subsequent changes in the manuscript, I hope there is now sufficient clarity to make that judgement.

*Major comment:• I find the current description of the methodology insufficient, especially considering the journal's modelling scope and the importance of the 'Prophet' procedure for this work. All that is said is that 'Prophet is a procedure for forecasting time-series data based on an additive model where non-linear trends are fit with yearly,weekly, and daily periodicity' and then the authors point towards a reference and some 'interpretive' parameters. This is simply not enough. A schematic would help and there has to be additional insight into how the algorithm works.*

**Response:** Apologies. I am happy to provide more detail by re-writing and expanding section 2 to better describe the workflow of our study and provide more details about the Prophet model as given in the official documentation and supporting peer reviewed paper. I will refer back to these modifications in the proceeding comments. I suggest the following additions that include the numerical framework Prophet is built on, how we include meteorological and traffic data and clarifying the standard approach for time-series cross validation. In our original manuscript, in section 2.1 we state that *Prophet is a procedure*

*for forecasting time series data based on an additive model where non-linear trends are fit with yearly, weekly, and daily periodicity. Developed for Facebook by* **?**, *it has found applications across various domains, not least driven by the underlying rationale to develop a modular regression model with interpretive parameters that can be intuitively adjusted by analysts with domain knowledge about the time series (e.g.,* **???**). I suggest adding the following text straight after this section:

Prophet belongs to a family of empirical models designed to forecast a variable as a function of time once model parame- ters have been optimised to historical observations. Whilst deterministic models are built around numerical representations of

[Figure]

**Figure 1.** Schematic illustrating an array of observations at one site, $y(t)$, and subsequent array of time periods the observations were made known processes (e.g. emissions, advection, oxidation etc), these empirical models rely on a time series of historical observations of the variable of interest which, for this study, is the concentration of $NO_2$. In our study, we have 5 years of observations, taken every hour, of the concentrations of $NO_2$ at 114 sites across the UK. These sites are described in more detail in section 2.3. Take the schematic provided in figure 1. This schematic represents an array of observations at one site, $y(t)$, and subsequent array of time periods the observations were made. The time between each observation, $\Delta t$, remains constant at one hour. If we fit the model to 3 years of hourly observations up to time $t_0$, the model is then able to predict the concentration of $NO_2$, thus $y(t)$, every hour from time $t_0 + \Delta t$. The user specifies how far into the future, thus how many hours, the model provides estimates for $NO_2$.

Ideally the model would also provide an estimation of forecast uncertainty at each point in time. Conceptually, the further away we move from time $t_0$, we might expect the error to increase, though this will depend on a number of factors we discuss shortly. In this study we have chosen one month of hourly predictions at all sites in order to evaluate model accuracy. It is of course important to understand whether the numerical architecture behind any time series forecasting technique is appropriate for the problem being studied. As noted in the Facebook research post [add ref], and paraphrased in the following bullet points, Prophet was originally designed for business forecast tasks which have any of the following characteristics:

– hourly, daily, or weekly observations with at least a few months of history

– strong multiple "human-scale" seasonalities: day of week and time of year

– important holidays that occur at irregular intervals that are known in advance

– historical trend changes, for instance due to product launches or logging changes

– trends that are non-linear growth curves, where a trend hits a natural limit or saturates

For forecasting atmospheric concentrations of $NO_2$ our chosen problem matches the first four. It is well known that NOx, which is the sum of NO and $NO_2$, has both natural (e.g., lighting and soil emissions) and anthropogenic (e.g., fossil fuel combustion and burning) sources and accurate predictions of its emission are critical to our understanding of ozone pollution and secondary organic aerosol formation. The concentration of $NO_2$ measured in the atmosphere varies as a function of emission, loss processes and meteorology, which results in a seasonally dependant concentration. Emissions of NOx vary seasonally due to changes in heating, burning and transport changes. Loss rates of NOx are dependent on meteorological conditions, photolysis rates and radical concentrations, primarily OH, all of which show seasonal dependencies. In the presence of sufficient levels of water vapour, a higher solar flux results in increased OH levels, which then can react with $NO_2$ for form $HNO_3$, the main terminal sink for NOx. Because of a relatively short chemical lifetime of around one day, NO2 regional distribution is strongly, but not solely, controlled by its local emission and thus traffic levels can have a huge impact on the measured concentrations.

In this study we use hourly observations from the AURN dataset described in section 2.3. Given the anthropogenic sources and meteorological factors that control $NO_2$, we expect that concentrations display diurnal to yearly variations according to diurnal sources and changes in environmental conditions. We also know that implementations of nationwide and local interventions such as changing the mix of vehicle types on the road or creation of inner city clean air zones have influenced the annual trends of $NO_2$ in the UK [add ref]. Figure 2 displays the auto-correlation and partial auto-correlation of $NO_2$ at an 'Urban Traffic' location in our dataset, specifically at roadside on Maryleborne Road in London [UK-AIR ID: UKA00315]. Figure 3 on the other hand displays the same information for a 'Rural Background', specifically Aston Hill in North Wales [UK-AIR ID: UKA00137]. Both figures illustrate the strength of a relationship with a single observation of $NO_2$ with observations at prior time steps. In this figure, the x-axis represents the number of lags from a given observation which in this case represents the number of hours. According to the characteristics of each site, we can see a clear diurnal correlation in the roadside location which reflects the contribution from the local traffic sources to measured concentrations of $NO_2$. Whilst both sites will be influenced by regional sources and variability in meteorological conditions, the rural background site displays a weaker diurnal pattern concomitant with the nature of local versus background sources at that site. As we state in section xx, we provide an archive of the analysis for all 144 sites.

With the above narrative in mind, before we provide an initial evaluation of using Prophet for forecasting $NO_2$, it is worthwhile providing a brief overview of the numerical architecture behind Prophet. Prophet is is an additive regression model with four main components as shown in equation 1 [add ref].

$$y(t) = g(t) + s(t) + h(t) + e(t) \tag{1}$$

where we have already defined $y(t)$ in our study as the concentration of $NO_2$. Variable $g(t)$ represents a trend term, $s(t)$ a seasonality term, $h(t)$ accounts for national holidays and $e(t)$ represents a noise term not accounted for by any of the previous terms. The user can specify a number of options before fitting this framework to historical data. The trend term can be represented as either a saturated growth model or by using the default linear model. The former would suit data that demonstrates potential for converging towards a known maximum point and might be best modelled using a logistic growth term. In this study we use the default linear model. Prophet also detects points where this rate has to change, referred to as change-point detection. Again, Prophet offers a default number of change-points or allows the user to specify dates where

[Figure]

**Figure 2.** Autocorrelation [top] and Partial autocorrelation for concentrations of $NO_2$ at Marylebone road, London, representative of an 'Urban Traffic' site described in section xx. The x-axis represents the number of lags from a given observation which in this case represents the number of hours. A light blue region represents a 95% confidence interval.

these have occurred. We have already briefly discussed the processes that dictate the concentrations of $NO_2$. With a deep interrogation and understanding of human enforced and natural events that might change the concentrations of $NO_2$ at any given site, the user might be able to specify change-point manually. For example, this might include identification of a traffic intervention. In this study we have used the default option but discuss potential improvements in section xx with regards to $NO_2$ and other pollutants. The seasonality component $s(t)$ accounts for periodic changes at the hourly, daily, weekly, monthly and yearly scales. Why is this useful? As already discussed, given the sources and controlling factors driving concentrations of $NO_2$ we know that human activity can exhibit a diurnal pattern. Likewise we know that changes in weather conditions occur across multiple scales. Prophet uses a Fourier series to represent periodic contributions, automatically detecting the frequency supplied in the data set being fit to. For example, if the user supplied data with a daily frequency, it would not make hourly forecasts.

However, given the influence meteorological conditions have on concentrations of pollutants, we can also use additional regressors to predict concentrations of $NO_2$ as a function of time and meteorological conditions. The extra regressor must be

[Figure]

**Figure 3.** Autocorrelation [top] and Partial autocorrelation for concentrations of $NO_2$ at Aston Hill, North Wales, representative of a 'Rural Background' site described in section xx. The x-axis represents the number of lags from a given observation which in this case represents the number of hours. A light blue region represents a 95% confidence interval known for both the history and for future dates. It thus must either be something that has known future values, or something that has separately been forecasted elsewhere. For example, one can also use as a regressor another time series that has been forecasted with a time series model [doc ref]. Figure xx displays a schematic illustrating an array of observations at one of our sites, $y(t)$, and subsequent arrays of meteorological factors (Wind speed, wind direction and temperature) and time periods all observations were made. In the previous example our forecast was based solely on a future period the user defined which, in our study, is every hour from a time $t_0$ highlighted in both figures 1 and 4. However, when fitting additional regressors we also need to provide an array of every additional metric (Wind speed, wind direction and temperature) at the same periods after $t_0$ and with the same temporal resolution.

The underlying numerics of adding extra regressors is that these are included in the linear component of the model, so the predictive time series depends on the extra regressor as either an additive or multiplicative factor. If a linear contribution is selected, equation 1 becomes:

$$y(t) = g(t) + s(t) + beta * z(t) + h(t) + e(t) \tag{2}$$

where $z(t)$ is the time series of the extra regressor and $beta$ is fit to the data. If additional regressors are included as a multiplicative factor, equation 1 becomes [ref doc]:

$$y(t) = g(t) * (1 + z(t) * beta) + s(t) + h(t) + e(t) \tag{3}$$

In code listing 1 we provide a snapshot example of importing Prophet into a Python script, configuring a default instance and then fitting to a time-series stored in a Pandas data-frame:

```python
**Import the Prophet framework**
from fbprophet import Prophet
import Pandas as pd

**Load a time-series data set of NO2, met and traffic data into a Pandas data-frame 'combined_df'>**

**Create a new data frame, 'train_dataset', that is used to train and then predict from a Prophet**
    instance
**The variables 'ds' and 'y' are required by the Prophet instance to identify the time and variable we**
    wish
**to predict respectively.**

train_dataset= pd.DataFrame()
train_dataset['ds'] = (pd.to_datetime(combined_df['Sdate']))
train_dataset['y']=combined_df['NO2']
```

In the previous section we outline how Prophet is able to include extra regressors in fitting the model. Using this approach we can include both meteorological data and traffic data for those sites that have it. However, there two alternative approaches we explore in this study. Whilst it has already been mentioned that the seasonality component $s(t)$ automatically accounts for periodic changes at the hourly, daily, weekly, monthly and yearly scales, the user can also specify a seasonality contribution that meets certain criteria. In our case, knowing that variable wind direction can result in distinctly different air masses and thus concentrations arriving at a site, we may also wish to go beyond the default daily seasonality and allow Prophet to fit contributions from different wind sectors in the training phase. In our data set we have wind direction given in degrees. In code listing 2 we expand our initial Pandas data-frame to include Boolean values for each wind sector.

```python
train_dataset['N'] = ((combined_df['wd'].values >= 348.75) & (combined_df['wd'].values < 11.25))
train_dataset['NNE'] = ((combined_df['wd'].values >= 11.25) & (combined_df['wd'].values < 33.75))
train_dataset['NE'] = ((combined_df['wd'].values >= 33.75) & (combined_df['wd'].values < 56.25))
train_dataset['ENE'] = ((combined_df['wd'].values >= 56.25) & (combined_df['wd'].values < 78.75))
train_dataset['E'] = ((combined_df['wd'].values >= 78.75) & (combined_df['wd'].values < 101.25))
train_dataset['ESE'] = ((combined_df['wd'].values >= 101.25) & (combined_df['wd'].values < 123.75))
train_dataset['SE'] = ((combined_df['wd'].values >= 123.75) & (combined_df['wd'].values < 146.25))
train_dataset['SSE'] = ((combined_df['wd'].values >= 146.25) & (combined_df['wd'].values < 168.75))
train_dataset['S'] = ((combined_df['wd'].values >= 168.75) & (combined_df['wd'].values < 191.25))
train_dataset['SSW'] = ((combined_df['wd'].values >= 191.25) & (combined_df['wd'].values < 213.75))
train_dataset['SW'] = ((combined_df['wd'].values >= 213.75) & (combined_df['wd'].values < 236.25))
```

```
train_dataset['WSW'] = ((combined_df['wd'].values >= 236.25) & (combined_df['wd'].values < 258.75))
train_dataset['W'] = ((combined_df['wd'].values >= 258.75) & (combined_df['wd'].values < 281.25))
train_dataset['WNW'] = ((combined_df['wd'].values >= 281.25) & (combined_df['wd'].values < 303.75))
train_dataset['NW'] = ((combined_df['wd'].values >= 303.75) & (combined_df['wd'].values < 326.25))
train_dataset['NNW'] = ((combined_df['wd'].values >= 326.25) & (combined_df['wd'].values < 348.75))
```

Following this, in code listing 3 we create an instance of the Prophet model but remove the single default daily seasonality contribution. In this study, this would include derivation of a 24 hour profile for NO2 that contributes to $s(t)$ in addition to a monthly and yearly contribution.

```
pro_regressor= Prophet(growth='linear', daily_seasonality=False)
```

In setting our own daily seasonality we can allow the model to generate a 'typical' diurnal profile under different wind sectors. In code listing 4, once an instance of Prophet has been created, we define a series of daily seasonality contributions that are fit when Boolean criteria present in our training data-frame, and defined by the variable $condition_name$, are met.

```
pro_regressor.add_seasonality(name='N',period=1,fourier_order=12,mode='additive', condition_name='N')
pro_regressor.add_seasonality(name='NNE',period=1,fourier_order=12,mode='additive', condition_name='NNE'
       )
pro_regressor.add_seasonality(name='NE',period=1,fourier_order=12,mode='additive', condition_name='NE')
pro_regressor.add_seasonality(name='ENE',period=1,fourier_order=12,mode='additive', condition_name='ENE'
       )
pro_regressor.add_seasonality(name='E',period=1,fourier_order=12,mode='additive', condition_name='E')
pro_regressor.add_seasonality(name='ESE',period=1,fourier_order=12,mode='additive', condition_name='ESE'
       )
pro_regressor.add_seasonality(name='SE',period=1,fourier_order=12,mode='additive', condition_name='SE')
pro_regressor.add_seasonality(name='SSE',period=1,fourier_order=12,mode='additive', condition_name='SSE'
       )
pro_regressor.add_seasonality(name='S',period=1,fourier_order=12,mode='additive', condition_name='S')
pro_regressor.add_seasonality(name='SSW',period=1,fourier_order=12,mode='additive', condition_name='SSW'
       )
pro_regressor.add_seasonality(name='SW',period=1,fourier_order=12,mode='additive', condition_name='SW')
pro_regressor.add_seasonality(name='WSW',period=1,fourier_order=12,mode='additive', condition_name='WSW'
       )
pro_regressor.add_seasonality(name='W',period=1,fourier_order=12,mode='additive', condition_name='W')
pro_regressor.add_seasonality(name='WNW',period=1,fourier_order=12,mode='additive', condition_name='WNW'
       )
pro_regressor.add_seasonality(name='NW',period=1,fourier_order=12,mode='additive', condition_name='NW')
pro_regressor.add_seasonality(name='NNW',period=1,fourier_order=12,mode='additive', condition_name='NNW'
       )
```

Please note that when specifying a manual seasonality, the user must also specify order of the Fourier series. Increasing the number of Fourier terms allows the seasonality to fit faster changing cycles, but can also lead to over-fitting [dpc - ref]. Whilst the non-user defined Fourier orders are left to the recommended default value, in this study we vary the Fourier order of our wind sector defined seasonality between 3, 5 and 10.

*Major comment:• I am also not entirely sure what is meant by 'rolling forecasts 30 days in advance' . Do I understand correctly, that you train and cross-validate on three years and then predict every single hour for the next 30 days in advance before moving the training interval 15 days forward in time? What are the predictors in this procedure? I assume it must be meteorological variables and NO2 concentrations of the previous hour, and lagged weekly, daily, and yearly values ? From the entire manuscript I could not fully comprehend what your predictions are actually based on.*

**Response:** In response to this comment I suggest adding a visual schematic of how this particular time-series forecasting technique works as per the previous response, whilst also including a new subsection that details the method for validation.

[Figure]

**Figure 4.** Schematic illustrating an array of observations at one site, $y(t)$, and subsequent arrays of meteorological factors [Wind speed, wind direction and temperature] and time periods the observations were made

The suggested text is given below which will now immediately follow the expanded section on the architecture of Prophet and the rationale for using this to forecast atmospheric concentrations of $NO_2$.

**2.3 Prophet evaluation:**

It is standard practice to evaluate model performance through the process of cross validation. Imagine we wish to predict a variable $y$ as a function of another variable $x$. After defining a training set from both $x$ and $y$, we likewise define a test set on which to evaluate a model performance. The ratio of data used from our entire dataset might be split 80%-20% between both the training and test set respectively. In a procedure often referred to as 'k-fold cross validation', the training set is split into k smaller sets and, through a series of k iterations, the resulting model is validated on the remaining part of the data. The performance reported by k-fold cross-validation is then the average of the values computed in the loop. The relative position of both the train and test set is typically shuffled according to a number of different strategies. For time series data, the procedure is slightly different. This is summarised in figure x, and we refer to reader to, for example, section 3.1.2.6.1 of the Sci-kit learn documentation for generic examples of time series validation [ref].

In this figure, the first array schematic highlights a period of observations that we fit the model to, before time $t_0$. Following this, there is a period beyond time $t_0$ that we predict values of our variable $y(t)$. In this hypothetical example, we predict values up to time $t_0 + 4\Delta t$ and, given we already have empirical observations of $y(t)$ in those time periods, we are able to evaluate the model performance. Using the previous discussion around a train and test set, we have likewise used a training set and validated our model against a test set. However, unlike the previous discussion, our model is used to forecast values beyond a set point. In this case, our training data always precedes our test data. In this case, we then move the location at which we stop fitting to

[Figure]

**Figure 5.** Schematic illustrating an array of observations at one site, $y(t)$, and subsequent arrays of meteorological factors [Wind speed, wind direction and temperature] and time periods the observations were made historical data in what is often referred to a 'rolling forecast'. For example, the first forecast is made by fitting to hourly data from 1st September 2016 00:00hrs - 1st September 2018 00:00hrs and then predicting concentrations between 1st September 2018 01:00hrs to 30th September 2018 01:00hrs. The second forecast is made by fitting to hourly data from 15th September 2016 00:00hrs - 15th October 2018 00:00hrs and then predicting concentrations between 15th October 2018 01:00hrs to 14th November 2018 01:00hrs. Comparing the predicted value at every hour in each forecast with observations provides us with the evaluation metrics. In our study, we fit to 3 years of historical observations and forecast concentrations over the next 30 days between Jan 1st 2018 and December 31st 2019. For each validation step, $t_0$ is increased by 15 days.

*General comment:My initial impression was that you actually try to predict NO2 30 days in advance at some point, which I assume cannot possibly be the case? Does, e.g.,the prediction at day 20 in the prediction period at noon, know of the true NO$_2$ value just one hour earlier? Please clarify these aspects.*

**Response:** Yes, this is correct. As I note in the previous response, a time-series forecast technique allows us to forecast values into the future based on trends in the past. In a similar way a regional forecast would not 'know of the true NO$_2$ value just one hour earlier', the Prophet model combines hourly to yearly contributions to forecast values at any given time based on model parameters optimised to historical data. I hope the suggested addition to the manuscript clarifies this point.

*General comment:Abstract l4-l6: could you be more quantitative or at least indicate somehow what"promising performance" means? When I read this for the first time, I was also confused what 'regional' sources are compared to 'non-local' sources and why BOTH should affect performance negatively. What would be a good setting in comparison to these two settings?* **Response:** With regards to the first point, a combination of absolute percentage deviation demonstrates that predictions from Prophet can be within 20of measured values at roadside locations, with errors increasing as the site being studied has a weaker diurnal signal. We have shown that we can include traffic data in the model fitting to capture the observed changes in $NO_2$ following significant changes in traffic levels. Combined with the ease of deployment when compared with e.g. a regional model, and the difficulties one might face in including a change in contributions from traffic variability, this shows promising performance. However, more work is needed to further evaluate the method, which would require a number of considerations we discuss at the end of the paper. With regards to the second, I agree this is perhaps somewhat confusing to combine regional and non-local as separate entities. I suggest removing the regional reference and providing more background of the nature of $NO_2$ in terms of sources and variability, as highlighted in the previous response. This hopefully helps clarify the common narrative around the definition of regional and local sources and map to the classification of site types also listed in Table 1.

*General comment:L10: I find the 'simplified approach of fitting to derived NO2-per-traffic volume' approach is introduced somewhat surprisingly and without context. Why is this used? Is that part of one of the models, as indicated by the word 'despite' at the beginning of the sentence? Please clarify.* **Response:** I suggest this is changed in the abstract. Specifically, the following change has been made: Using a relatively simply approach to incorporate traffic volume into the model fitting and thus forecastDespite the simplified approach of fitting to derived NO2-per-traffic volume over a 5 year period, trends in absolute $NO_2$ reductions and diurnal profiles were captured well at Manchester Piccadilly.

*General comment:L16: 'effective and simple' relative to what?* **Response:** Relative to setting up and deploying a regional model on a high-performance cluster to predict values at individual sites which would need further modifications to incorporate the significant changes in traffic volume. We do, however, demonstrate the value in combining the two different approaches.

     *General comment:L21: clarify the timescale of predictions you are interested in. Minutes? Simul-taneous (from other mea-surements)? Hours? Days? As I said, I am not entirely sure yet if you actually intend to predict just the update in NO2*

*concentrations from one hour to the next, or over longer time intervals.* **Response:** I hope the additions now made to the manuscript now clarify this.

     *Specific comment:L30: Could you give examples of what those challenges are?* **Response:** Yes I am happy to do so. I suggest the following additional sentence is now added: For example, a regional model typically requires the use of a high-performance computing [HPC] facility. Access to such facilities can be a heterogeneous issue, from both a resource and support perspective.

Whilst such models are built around robust numerical representations of known processes that cover emissions, advection, deposition and so on, it can be difficult to embed processes that may be important at hyper-local scales such as variable traffic flows. This is largely driven by an existing computational complexity that likewise can dictate the time-to-solution. Of course, each model has variable capabilities with regards to processes captured and the level of computational optimisation that can be achieved. Time-series forecasting methods can he developed and deployed on personal computing devices, again depending on the level of complexity required. For example, methods built around deep learning architectures may require access to a minimum specification of Graphical Processing Unit [GPU] during model training. In this paper, we use a statistical package that can be trained and deployed on a personal computing device.

*Specific comment:L43: packageS such as Scikit-Learn and bracket typos. Keras has no official citation?* **Response:** This has now been corrected to the correct reference style, and I have added a reference for Keras.

*Specific comment:L51: define what you mean by 'local' – point/site measurements I suppose?* **Response:** Yes this is a good point, local is too vague a definition. I suggest replacing this with the following: Overall the Prophet model offers a relatively effective and simple way to make predictions about $NO_2$ at specific points/sites.

*L61: here you mention for the first time that you aim to predict NO2 concentrations a month in advance (which could be*
*understood in different ways). I think this should be clarified earlier on, even in the abstract.***Response:** Yes I agree. Section 2.1 has now been re-written to incorporate the text provided in response to your earlier comments and, with the new schematic, clarifies the hourly resolution of the forecast.

*General comments on section 2.1: given how central the Prophet model is to your paper, I would want to see a more detailed*
*description of what happens 'under the hood'. Currently, and given that I don't know the method, it seems like a black box to me, which is impression you would certainly want to avoid. A graphic/schematic would help, too. Why should the Prophet model be advantageous over for example LSTMs? Explaining the modeling process would be really important to make the paper more interesting and more accessible.* **Response:**. I hope the new additions address this concern. With regards to any advantages over LSTMs, this is not something we can demonstrate here. In the original paper, which was submitted as a model*
evaluation paper, there was no reference to any suggested advantage over LSTMs. To make any comparison would require us to design and evaluate a relevant architecture for a given LSTM. It is not clear whether each site studied in this project would warrant a seperate LSTM design and would for sure be an interesting case study. The framework that Prophet is based on is specified in the original documentation, and now drawn out in the revised manuscript. The code is open source and the project repository and archive provided with the original and new manuscript enable readers to replicate our results.

*L78 typo processes***Response:**.This has been corrected.

*L85: TNO database not defined yet. Maybe link website if appropriate?* **Response:**. Sure, this has now been added.

*L.89-91: without context this doesn't make sense to me. If there are 2-months periods, what happened to Jan/Feb 2020? I assume you treat 2020 separately due to the lock-down? Maybe worth pointing out already at this stage.* **Response:**. This is
clarifying that, for our EMEP simulations, predictions were provided at hourly resolution over 2 months. I suggest the paper now reads as follows: In this study EMEP was used to provide hourly predictions in blocks of 2 month periods over 2016 to end of 2019. Each 2 month block was preceded by a 7-day spin-up period to initialise the chemical fields. The 3-month simulation for 2020 (March-May) was run as a single period, with a 7-day spin-up period at the end of Feb 2020.

*L.103: make clearer how the different time periods align somewhere in the manuscript (certain years are used for training, others for prediction and EMEP modelling, I suppose, others again to test the effects of the lock-down?) L.131: 3. Results. By this point in the manuscript, I am still not sure how you predict NO2, but you already start presenting results. What are*
*the variables you use as predictors (and at what time lags)? How do you cross-validate to avoid overfitting (you mention that something is done in section 2.1, but I feel this is insufficient for a modelling paper)?* **Response:**. I hope the new paper structure and visualisations now clarify this.

*How do you evaluate model skill? Surely the Pearson correlations shown in Figure 1b are performed on test data and not on training data, i.e. the sequential predictions on predicted months? Your current manuscript simply does not describe this in*
*sufficient detail and clarity in my opinion and I find it difficult to judge the skill of your approach as a result.* **Response:** I hope the previous responses help clarify the entire workflow in fitting the Prophet model and making forecasts.

*Figure 1b: Please compare these correlations to time series where you simply prescribe a seasonal cycle (smoothed over several days) or a constant value that is representative of the true annual mean observed values. This would provide a much clearer impression of how much better your predictions actually are compared to very basic models. Urban sites I would*
*expect to show less relative (in %) seasonal variation so that this might explain your smaller error there. Furthermore, try the R2-score (coefficient of determination) rather than just Pearson correlation. The former should be better suited to compare time series of this kind because it does not just consider variance but also magnitude of the prediction error.* **Response:** This is a good idea and will be included in a revised manuscript.

*L153: I am sceptical that the EMEP model is a useful benchmark here. It simply seems to have a high bias, but how about*
*an empirical model that simply has no mean bias (for instance the seasonal cycle model mention above, derived from actual observations)?* **Response:** We do not use EMEP as a benchmark nor do we state this in the document. EMEP is used for comparison and interrogation of periosd where Prophet forecasts performed poorly, especially during the first UK COVID19 lockdown.

*L175: At this point it is still not clear to me which variables were actually included in the predictions of results section 3.1. . . /*
*Figure 7: relabel y-axis with logarithmic scale I suggest. / L185: I think this approach requires a more detailed explanation. How is this done exactly with Prophet?* **Response:** I hope the clarified workflow now helps. I refer the reviewer back to the previous response regarding the methdology.

---

## Author Comment (AC2) · 1 Mar 2021

Dear colleague.

Thank you for your review received 24/12/20. I am of course very happy to respond to the points raised and hope the added justification and clarity of approach significantly improves the manuscript.

*Overall Comments: The analysis needs to be extended as the whole forecast workflow to understand the human impact on NO2 concentrations and forecasting seems to be procrustean to Facebook's Prophet model.The work that needs to be done for this contribution is major, as relevant forecast diagnostics need to be included, regardless of whether they are implemented by Facebook or not.The novelty of the paper is the workflow that involves resolving multiple data sources, that is then used in forecasting and clustering.*

*The workflow does not exist in the write-up (methodology section). In addition, exploratory data analysis to drive the type of model used (referred to as vanilla without any mathematical description) needs to be provided. Authors need to provide motivations behind models they select and this is not possible without exploratory data analysis. Missing are general approaches such as partial autocorrelation analysis of time series, methods used for resolving different spatial scales of input data,the type of forecast method used (including the trend and seasonality model), and why Prophet model is chosen.*
**Response:** I hope the additional material added to the manuscript, in response to the specific points also repeated below, are sufficient. It is unclear why the reviewer is referring to '*different spatial scales of the input data*'. All of the environmental and mobility data are from fixed sites, or point source measurements. The domain setup of the EMEP model is described. Whilst we add some background text on sources and behaviour of NO2 for ease of reading and to better understand the rationale, the definition of the AURN site type is pre-defined by local authorities in order to meet the requirements of the review and assessment process for Local Air Quality Management.

*A new section on exploratory data analysis with concrete mathematical motivations to use a specific model is a must for this paper. Prophet model needs to be explained in the context of the problem.*
**Response**: The new manuscript will include a description of the Prophet framework. It will also include a brief background summary of the known sources and sinks, thus seasonal nature of NO2 for the non-atmospheric reader. It will also include a presentation of auto-correlation plots from a subset of the sites studied, linking to an archive of plots for all sites.

*Comments on Sections:*
*2.1 needs to be extended. In its current form this section is not informative about the model being used. The prophet model has many models, pre and post processing steps. Models used in the study need to be clearly discussed.*
**Response.** Yes I agree, this was an oversight in referring to the Prophet documentation alone. I have added a description of the model used.

2.2 Similar problem as 2.1 This section contains an overly abridged description of the method followed by details regarding application. This section needs to be rewritten.The methodology requires a workflow section that describes the overall workflow built for this study.
**Response**: This has now been included.

Table 1 shows differing spatial resolutions, authors need to clearly present how this disparity is resolved (interpolation, sampling, etc.)
**Response:** It is unclear what additional analysis we could offer, or is to be expected, regarding Table 1, but perhaps additional text may help the reader. Table 1 lists air quality measurement sites managed by the automatic Urban and Rural Network (AURN) in the UK. The network has grown in response to legislative, scientific, technical and policy requirements over time. The classifications provided in Table 1 are not dictated by the authors of this manuscript, rather have been assigned by local authorities in order to meet the requirements of the review and assessment process for Local Air Quality Management. Indeed, In order to ensure the UK's compliance with the EU air quality Directives, the specific categories listed in table 1 are used and are based on known sources and dispersion of key pollutants. In the revised manuscript I suggest the following text is added in section 2:

"*Table 1 lists air quality measurement sites managed by the automatic Urban and Rural Network (AURN) in the UK. The network has grown in response to legislative, scientific, technical and policy requirements over time. The classifications provided in Table 1 are not dictated by the authors of this manuscript, rather have been assigned by local authorities in order to meet the requirements of the review and assessment process for Local Air Quality Management. Indeed, In order to ensure the UK's compliance with the EU air quality Directives, the specific categories listed in table 1 are used and are based on known sources and dispersion of key pollutants.*"'

2.4 This section contains no information on data. If the only purpose of this section is state the changes to mobility patterns after March 12th, this can be done in the section on data sources.
**Response**: I agree. This has been changed.

The prophet model is a general additive model for time series, where seasonanlity, trend, and cyclicity are linearly superimposed. There is no motivation in the paper why such additive model will be appropriate for NO2 concentrations. It is customary to provide at least a partial auto corelation function plot of the signal in question and performing statistical tests on these components to understand whether an additive model would be appropriate to start with.
**Response:** The new manuscript will include a brief background summary of the known sources and sinks, thus seasonal nature of NO2 for the non-atmospheric reader. It will also include a presentation of auto-correlation plots from a subset of the sites studied, linking to an archive of plots for all sites.

Specific Comments:
Line 2: Missing comma after In this study
**Response:** This has been corrected.

Line 12: "the nature of local traffic": artificial constructs do not possess natural qualities.
**Response**. This is irrelevant. To comment on the nature of something is to comment on its basic or inherent features, character, or qualities.

If you mean historic patterns, please write so.
**Response**: This has been clarified.

Line 13: Why is HGV abbreviation in square brackets?
**Response**: The bracket format is now consistent throughout the manuscript.

Line 16: Overall missing comma
**Response:** This has been corrected.

Line 56-60 belongs in the introduction section.
**Response**: The introduction section has been restructured to accommodate the requests on background and rationale.

Line 62: "The internal cross-validation methods provided by Prophet are used to arrive at a set of performance metrics applied across all sites." This is not specific enough, you need a subsection under methodology for diagnostics and discuss why they are relevant.
**Response**: The introduction to the numerical framework Prophet is based on, and the use of cross-validation for time-series forecasting, is now explained in a new re-written section 2.

Line 63-64: 'this process includes fitting to historical data over a specified period", does this imply you are using an autoregressive forecasting method. If so, this does not imply novelty as these methods have been studied intensively for a long time.
**Response**: There is no claim the chosen method is novel anywhere in the manuscript. This paper was submitted as a model evaluation study.

Line 65 - 74: These descriptions have nothing to do with the Prophet model but the problem definition. Please distinguish the method from the application.
**Response**: As the reviewer has quite rightly raised in the preceding comments, it is important to clarify the rationale behind any given method according to the nature of the problem being solved. This is now improved significantly in a revised manuscript.

Line 82-91: Here application is discussed, this section is not related to the method.EMEP model is an integral piece of this work, please describe the system being solved with this model here.
**Response:** In the revised manuscript we now clarify how EMEP is used to simulate the concentrations of key pollutants, including NO2, across the UK with higher resolution nests focused over Manchester.

'Line 94: need to be carried over to notes or the appendix
**Response:** This has been changed.

Line 98: "meteorological data on wind speed, direction and temperature provided by the UK Met Office." Please elaborate which model, data product, whether it is reanaly-sis data. In addition, please introduce

**Response**: This is a good point and will be added to the manuscript.

Line 116: "this method captures between 8 % – 15 % of vehicles making a journey between two sensors,". Does this mean the traffic data captures up to 15% of actual traffic. If so the sampling bias, and portion of the traffic needs to be discussed in extenso.

**Response**: We do not have access to the statistics that would allow us to discuss the sampling bias in full [in extenso]. However we can ensure that, despite an open presentation on the assumptions made, this should be considered in how any further study builds on this.

Line 156-157: "The multi-modal behaviour of percentage deviation by site type may be indicative of local interventions not captured by the default change-points used during the fitting process." This seems to be a speculation rather than a result. This can be an important point to bring up in the discussions.

**Response**: I agree. With a restructure in how the model results are presented, the site type variability is now part of a wider discussion.

Line 158 - 160: "One can alter the weighting given to such changes. A manual analysis on an individual site level might identify significant changes in local activity that would be expected to change the seasonality in measured NO2 and thus define change-points that need to be captured during the fitting process." This is a use tip aboutProphet, I don't think this relates to results.

**Response**: This is no longer included in the manuscript.

Line 164: Vanilla prophet is not discussed methodologically. This must be expanded in the methodology section.

**Response**: This has now been clarified in a rewrite of the methodologies used.

Line 169: Improvement in forecast from 18% to 10% deviation can be quite misleading. Authors need a transform bias correction scheme for the Yeo-Johnson transform to address this. Please see the seminal work of Beauchamp and Olson (1973) on pitfalls of using transformations. Secondly, a transformation is applied to data without being introduced in the methodology. There is no discussion of why this transformation is chosen over Box-Cox transform or a log transform.

**Response**: The new manuscript now includes a clear overview of the pre-processing steps used. In this instance, since we were transforming a log product, we used the Yeo-Johnson scheme.

Line 220: HYSPLIT model is introduced for the first time under results. This is a major part of the method yet not discussed in methodology.

**Response**: This has now been included in the revised manuscript.

Line 225: Ward's Method is introduced for the first time in results without discussion under methodology

**Response**: Apologies. This is standard practice. The purpose of back trajectory analysis is to confirm the arrival of different types of air masses at a given site. Whilst the specific clustering of trajectories will change with choice of linkage scheme, the usefulness of the results presented here was to confirm the change in air masses that coincided with significant increases in NO2 even during lockdown conditions.

Line 227- 229: "Discrepancies between forecasts that incorporate traffic and measuredvalues could arise from errors associated with a number of factors, but data on vehicle traffic type suggests this could also be due to an increase in the ratio of Heavy GoodsVehicles [HGVs]." This statement is not possible to make with current metrics and the level of exploratory analysis of variables. Bias residuals, in addition to very common forecast metrics such as MAPE and MASE are missing in the analysis. If NO2 concentrations have strong seasonality, then observing bimodal deviance (such Figure 1 is not surprising, that is just an indication that there are two regimes where model does significantly bad compared to others).

**Response**: Numerous studies of course confirm that different vehicles emit significantly different levels of NO2. This is core to discussions around clean-air zones and selective filtering schemes in urban areas. True it is not possible to statistically confirm this, but the evidence on a change in the nature of the local traffic could be significant. The new manuscript restructures the presentation of model performance.

Figure 4: On average, the model consistently over-estimates measured values. This needs to be discussed.

**Response**: This will now be included.